# GRAMIAN MULTIMODAL REPRESENTATION LEARNING AND ALIGNMENT

**Giordano Cicchetti**\*, **Eleonora Grassucci**\*, **Luigi Sigillo**, **Danilo Comminiello**
Dept. of Information Engineering, Electronics, and Telecomm., Sapienza University of Rome, Italy
`{name.surname}@uniroma1.it`

## ABSTRACT

Human perception integrates multiple modalities—such as vision, hearing, and language—into a unified understanding of the surrounding reality. While recent multimodal models have achieved significant progress by aligning pairs of modalities via contrastive learning, their solutions are unsuitable when scaling to multiple modalities. These models typically align each modality to a designated anchor without ensuring the alignment of all modalities with each other, leading to suboptimal performance in tasks requiring a joint understanding of multiple modalities. In this paper, we structurally rethink the pairwise conventional approach to multimodal learning and we present the novel Gramian Representation Alignment Measure (GRAM), which overcomes the above-mentioned limitations. GRAM learns and then aligns $n$ modalities directly in the higher-dimensional space in which modality embeddings lie by minimizing the Gramian volume of the $k$-dimensional parallelotope spanned by the modality vectors, ensuring the geometric alignment of all modalities simultaneously. GRAM can replace cosine similarity in any downstream method, holding for 2 to $n$ modalities and providing more meaningful alignment with respect to previous similarity measures. The novel GRAM-based contrastive loss function enhances the alignment of multimodal models in the higher-dimensional embedding space, leading to new state-of-the-art performance in downstream tasks such as video-audio-text retrieval and audio-video classification. The project page, the code and the pretrained models are available at `https://ispamm.github.io/GRAM/`.

## 1 INTRODUCTION

Humans naturally process and integrate signals from multiple sensory modalities, such as sounds and visual inputs, to form a coherent understanding of the world around them. Inspired by this, foundational models have attempted to replicate this capability by aligning pairs of modalities, such as vision and language, through contrastive learning techniques. One of the most significant contributions in this domain was CLIP Radford et al. (2021), which used a contrastive loss to align image and text representations based on cosine similarity. CLIP has shaped the current approach to multimodal learning, and every subsequent model relies on the same contrastive-pairs fashion, even in the case of models involving more than two modalities, such as ImageBind Girdhar et al. (2023), VAST Chen et al. (2023c), and LanguageBind Zhu et al. (2024).

However, these models suffer from critical limitations, as they typically involve cosine similarity to align each modality to a chosen anchor modality (e.g., images, text, or audio) Sirnam et al. (2023); Girdhar et al. (2023), without ensuring consistent alignment between non-anchor modalities. This approach can be insufficient for tasks that require cross-modal understanding beyond pairs. For instance, aligning video and audio separately to a common textual anchor does not guarantee that the video and audio themselves are well-aligned. Consequently, models often fail in complex, real-world scenarios where interactions between multiple modalities are crucial. A notable example is video-text retrieval, where audio can be essential for correctly interpreting the scene. State-of-the-art (SOTA) models often neglect this, leading to suboptimal performance when audio is a critical factor Yoon et al. (2023), as we show in Fig. 6 of Appendix B. Due to these inherent limitations,

---

\*These authors contributed equally to this work.

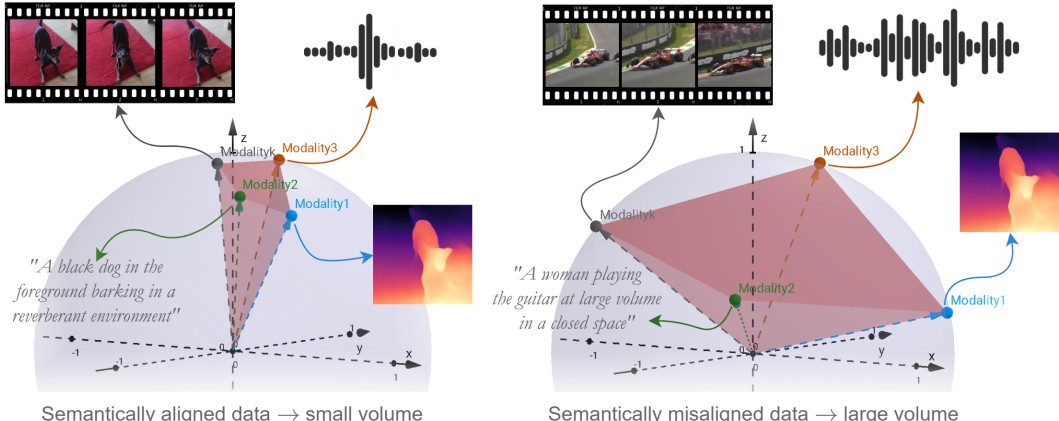

Figure 1: Visualization of the GRAM intuition: on the left, embedding vectors from semantically aligned multimodal data build a parallelotope with a small volume. On the right, where modalities are not aligned with each other, the formed parallelotope has a large volume.

the current trajectory of multimodal learning is reaching a plateau, with the sole possible research direction of scaling up models and datasets bringing negligible improvements.

This paper presents a novel alignment measure that fundamentally rethinks the alignment process of multimodal models, by directly operating in the higher-dimensional space spanned by the modality embeddings. The proposed measure can replace the cosine similarity in any downstream method providing more meaningful information about the embedding semantics, especially in the case of multiple modalities. Together with the origin of the axis, such $k$ embeddings of dimension $n$ define the edges of a $k$-dimensional parallelotope in $\mathbb{R}^n$, whose volume is strictly related to the alignment of the modalities as Fig. 1 shows. Indeed, the lower the volume, the closer the modality vectors are, meaning they are well-aligned. The Gramian Representation Alignment Measure (GRAM) exploits this intuition to provide meaningful information on the semantic alignment of more modalities in the higher-dimensional space in which they lie. Conversely to the conventional cosine similarity, the proposed GRAM evaluates the alignment of more modalities without requiring to compare pairs of them, therefore preserving the rich semantic relations of the multimodal space. Moreover, it is well-defined for 2 to $n$ modalities, making it extremely flexible and adaptable to any real-world scenario and task. Further, we introduce a novel volume-based contrastive learning loss function, which leverages the recently introduced GRAM to lead multimodal models in shaping a unified and aligned embedding space.

We experimentally show how rethinking the multimodal alignment process with GRAM brings a consistent improvement to the state of the art in downstream tasks, and that our intuition that more modalities altogether provide richer semantic information is validated. Specifically, the multimodal model pretrained with the GRAM contrastive loss outperforms SOTA models by 5 to 10 points in tasks such as video-text and audio-text retrieval or video classification in several datasets without any architectural modification and with the same number of network parameters. These results prove once again that GRAM better models the world multimodality with respect to the conventional pairwise alignment process.

Our main contributions can be summarized as follows:

- We introduce GRAM, a **novel similarity measure** for multimodal representation learning and alignment based on the volume computation of the parallelotope spanned by all the modality vectors. The proposed measure can replace cosine similarity in any downstream method, providing richer and more informative semantics on modality representations by guaranteeing geometric alignment between $n$ modalities simultaneously.

- We introduce a **novel GRAM-based contrastive loss** function that can be involved to pretrain or finetune any previous existing multimodal model, building a unified and aligned embedding space.

- We show that our measure also serves as a **quantitative metrics** for the performance of past and future multimodal models, providing unique insights on latent feature alignment.

- We provide the proof of the extension of GRAM from 2 **up to $n$ modalities**, mathematically proving, and experimentally showing, that GRAM findings hold for $n$ modalities, making it extremely generalizable.

- We show how multimodal models trained with the proposed contrastive loss outperform SOTA models, bringing **improvements in several downstream tasks** such as video-audio-text retrieval and audio-video classification.

## 2 RELATED WORK

**Two-modal Alignment.** CLIP Radford et al. (2021) originally revolutionized multimodal learning by providing a foundational model that effectively aligns two modalities, specifically images and text. This work has inspired subsequent advancements aimed at enhancing performance and alignment quality across modalities Uesaka et al. (2024); Ilharco et al. (2021); Zhai et al. (2023). Additionally, CLIP architecture has been adapted to facilitate alignment between other modality pairs, such as audio and text in CLAP Elizalde et al. (2023), video and text in CLIP4Clip Luo et al. (2021), and point clouds and text in PointCLIP Zhang et al. (2021). These models commonly rely on contrastive learning strategies, which push dissimilar embeddings apart while drawing similar vectors closer together, providing a robust method for learning cross-modal representations.

**Multimodal Alignment.** Recent research has extended beyond two-modal alignment to capture the more complex multimodal nature of real-world data Lyu et al. (2024). For instance, CLIP4VLA Ruan et al. (2023) integrates audio within the CLIP framework, aligning video, audio, and text in pairs, using audio embeddings as a central anchor. Building on this, ImageBind Girdhar et al. (2023) offers a pre-trained multimodal framework encompassing modalities like depth and infrared, positioning the image modality as the bridge across modalities. In contrast, LanguageBind Zhu et al. (2024) demonstrates that text, rather than images, is a more effective anchor for multimodal integration. Alongside these innovations, other models such as VALOR Chen et al. (2023b), VAST Chen et al. (2023c), mPLUG-2 Xu et al. (2023), VideoPrism Zhao et al. (2024), and InternVideo2 Wang et al. (2024b), among others, either propose large pretraining datasets or introduce architectural modifications that further optimize performance. Among these, VAST proposes to fuse multiple modalities by an MLP layer and then involve conventional contrastive losses between such fused omni-modality embedding and the anchor one. Despite the lack of interpretability of this solution, at least VAST builds the first attempt towards a multimodal-like space.

Despite their contributions, these approaches still operate within a constrained lower-dimensional space, often relying on cosine similarity on a 2D plane defined by two modalities or using fusion strategies to combine multiple modalities. As a result, these models fall short in fully exploiting the rich, high-dimensional information inherent in multimodal data, which is crucial for addressing more complex downstream tasks.

**Gram Matrix in Deep Learning.** The Gram matrix properties have been leveraged in deep learning models for better defining matrix theory behind these models Pennington & Worah (2017), or to improve performance in different downstream applications. Examples are learning rotation-invariant point cloud representations Xu et al. (2021), sound event detection Neto et al. (2021), neural style transfer Li et al. (2017); Friedrich & Menzel (2019), and domain adaptation Nejjar et al. (2023). In addition and interestingly, the Gram matrix has also been proved to bring some insights on GAN representations Seddik et al. (2020).

## 3 GRAMIAN REPRESENTATION ALIGNMENT MEASURE

Our goal is to achieve comprehensive geometric learning and alignment of multiple modality representations within their inherent high-dimensional spaces. Unlike traditional pairwise alignment methods, GRAM advances beyond pairwise limitations to introduce a novel strategy for modeling and interpreting latent multimodal representations in an integrated manner. By jointly learning and then aligning these modalities, our method exploits the full potential of multimodal models, while maintaining a high degree of geometric interpretability. The GRAM measure can replace cosine

similarity in any downstream method bringing several advantages such as enhancing model performance and providing meaningful information on latent representation alignment.

## 3.1 PRELIMINARIES

Multimodal representation learning aims to learn latent representations from co-occurrent data modalities. The $i$-th modality is encoded from its own domain in an $n$-dimensional latent representation using an encoding function $e_i : M_i \to \mathbf{M}_i$, with $\mathbf{M}_i \in \mathbb{R}^n$. A representative example may be the video-audio-text representation where we have a tri-modal representation with three encoding functions: $e_V : V \to \mathbf{V}$ as visual encoder, $e_A : A \to \mathbf{A}$ as audio encoder, and $e_T : T \to \mathbf{T}$ as text encoder. Here, $V \in \mathbb{R}^{N_f \times C \times W \times H}$ is the visual domain assuming videos with $N_f$ frames each one with $C$ channels and a resolution of $W \times H$, $A \in \mathbb{R}^{N_c \times N_s}$ is the audio domain assuming audio with $N_s$ samples for each $N_c$ channels, and $T = [t_1, \dots t_M]$ is the textual domain composed of a set of tokens $t_i \in Vocabulary$, $\mathbf{V}, \mathbf{A}, \mathbf{T} \in \mathbb{R}^n$.

Usually, the similarity between two modalities $\mathbf{M}_i, \mathbf{M}_j$ is obtained by computing the cosine of the angle $\theta_{ij}$ between their two representations:

$$\cos(\theta_{ij}) = \frac{\langle \mathbf{M}_i, \mathbf{M}_j \rangle}{||\mathbf{M}_i|| \cdot ||\mathbf{M}_j||} \tag{1}$$

where $\langle \mathbf{M}_i, \mathbf{M}_j \rangle$ is the dot product between modality $\mathbf{M}_i$ and modality $\mathbf{M}_j$, and $||\mathbf{M}_i||$ is the norm of $\mathbf{M}_i$.

Since cosine similarity is not defined for three or more vectors altogether, the classical approach is to select one modality as bridge modality and align all the remaining N-1 modalities to the first one couple-wise. In this scenario whatever type of contrastive loss could be used, the most common is the one introduced by Radford *et al.* in CLIP Radford et al. (2021):

$$\mathcal{L}_{M2A} = -\frac{1}{B} \sum_{i=1}^{B} \log \frac{\exp(\mathbf{m}_i^\top \mathbf{a}_i / \tau)}{\sum_{j=1}^{B} \exp(\mathbf{m}_i^\top \mathbf{a}_j / \tau)}, \mathcal{L}_{A2M} = -\frac{1}{B} \sum_{i=1}^{B} \log \frac{\exp(\mathbf{a}_i^\top \mathbf{m}_i / \tau)}{\sum_{j=1}^{B} \exp(\mathbf{a}_i^\top \mathbf{m}_j / \tau)} \tag{2}$$

where $m_i$ is the normalized embedding of the $i$-th modality data and $a_i$ is the normalized embedding of the bridge modality and $B$ is the batch size parameter.

At inference time, the problem becomes even more apparent since conventional methods do not possess a direct mathematical formulation to compute similarity among three or more embedding vectors. Therefore, they rely on just two modalities or develop suboptimal neural fusing strategies. For instance, LanguageBind Zhu et al. (2024) attempts to linearly combine two modalities and then compute similarity with the third one, while methods like UMT Liu et al. (2022), m-PLUG2 Xu et al. (2023) and VAST Chen et al. (2023c) introduce layers that fuse two or more modality embeddings before computing cosine similarity with the remaining one.

## 3.2 VOLUME OF THE PARALLELOTOPE SPANNED BY THE MODALITY VECTORS

A generic embedding vector $\mathbf{v}$ with dimension $n$ is a vector in the higher-dimensional space $\mathbb{R}^n$, whose components indicate its extremity. In the case of various modalities encoded with the correspondent encoding functions, multiple embeddings vectors $\mathbf{v}_1, \dots, \mathbf{v}_k$ lie in the higher-dimensional space $\mathbb{R}^n$, i.e. $\mathbf{v}_i \in \mathbb{R}^n, \forall i \in [1, k]$. The aim of whatever multimodal representation framework is to ideally have correlated embeddings near to each other and uncorrelated embeddings far away, so as to have an aligned and meaningful embedding space. In this paper, we argue that a measure of the relationship among the vectors in the hyperdimensional space can be given by the volume of the $k$-dimensional parallelotope with these vectors as sides. Recalling that the vectors are normalized to have a unitary norm, their extremities lie on the surface of the hypersphere with a radius equal to 1, as shown in Fig. 2. The way to obtain the volume of whatever type of $k$-dimensional parallelotope is computing the determinant of the Gram matrix $\mathbf{G}$ of the vectors Gantmacher (1959), as we show in this Section.

**Definition 1: Gram Matrix.** Let $\mathbf{v}_1, \dots, \mathbf{v}_k$ be vectors in $\mathbb{R}^n$, these points can be arranged as columns in a matrix $\mathbf{A} = (\mathbf{v}_1, \dots, \mathbf{v}_k)$. Then, the Gram matrix $\mathbf{G}(\mathbf{v}_1, \dots, \mathbf{v}_k) \in \mathbb{R}^{k \times k}$ is defined:

$$\mathbf{G}(\mathbf{v}_1, \ldots, \mathbf{v}_k) = \mathbf{A}^\top \mathbf{A} = \begin{bmatrix} \langle \mathbf{v}_1, \mathbf{v}_1 \rangle & \langle \mathbf{v}_1, \mathbf{v}_2 \rangle & \cdots & \langle \mathbf{v}_1, \mathbf{v}_k \rangle \\ \langle \mathbf{v}_2, \mathbf{v}_1 \rangle & \langle \mathbf{v}_2, \mathbf{v}_2 \rangle & \cdots & \langle \mathbf{v}_2, \mathbf{v}_k \rangle \\ \vdots & \vdots & \ddots & \vdots \\ \langle \mathbf{v}_k, \mathbf{v}_1 \rangle & \langle \mathbf{v}_k, \mathbf{v}_2 \rangle & \cdots & \langle \mathbf{v}_k, \mathbf{v}_k \rangle \end{bmatrix}. \tag{3}$$

***Theorem 1*: Volume of the $k$-dimensional parallelotope.** Given $\mathbf{v}_1, \ldots, \mathbf{v}_k$ be vectors in $\mathbb{R}^n$ forming a $k$-dimensional parallelotope, and the Gram matrix $\mathbf{G} \in \mathbb{R}^{k \times k}$, then its determinant, also called the Gramian, is the square of the volume of the $k$-dimensional parallelotope formed by the vectors Gantmacher (1959):

$$\text{Vol}(\mathbf{v}_1, \ldots, \mathbf{v}_k) = \sqrt{\det \mathbf{G}(\mathbf{v}_1, \ldots, \mathbf{v}_k)}. \tag{4}$$

Note that $\det \mathbf{G} = \det \mathbf{A}^\top \mathbf{A} = \det \mathbf{A} \cdot \det \mathbf{A} = |\det \mathbf{A}|^2 \geq 0$, so the square root in Eq. 4 is well-posed.

*Proof.* See Appendix A.

Interestingly, in the case of $k < n$, i.e., when the number of modality vectors $k$ is lower than the dimension of the ambient space $n$, the Gramian provides us a reliable computation for the volume of the $k$-parallelotope spanned by those vectors, yielding crucial information on the alignment of the modalities. Therefore, the Gramian computation of the volume is naturally and easily extended to the number of modalities that we want to use in our framework, from 2 up to $n$, where the latter is the dimensionality of the latent space. In the case when $k > n$ the vectors are linearly dependent, so the volume of the $k$-parallelotope is still positive but equal to zero and the Gramian is not informative. However, in current neural embedding models, $k \ll n$, so the findings hold for the largest part of real-world scenarios known up to now.

## 3.3 GRAM: VOLUME AS A MEASURE OF MODALITY VECTORS ALIGNMENT

The notion of volume in hyperdimensional spaces offers an intuitive approach for understanding the geometric alignment of vectors, which are the representation of data from various modalities. Specifically, the volume spanned by a set of vectors provides a measure of their proximity in the hyperdimensional space. A smaller volume indicates that the vectors are closely aligned, suggesting a close semantic relation between the underlying data. Conversely, a larger volume suggests that the vectors—and, by extension, the input data they represent—are less correlated or potentially opposite. Our Gramian Representation Alignment Measure relies on this crucial intuition to fully exploit the richness of the semantic data. This novel methodology facilitates the discovery of interrelationships between sub-modalities by computing the volume of the $k$-dimensional parallelotope formed by any subset of vectors. Interestingly, the GRAM computation is also efficient as it relies on computing the determinant of a $k \times k$ matrix, where, in the largest part of real-world scenarios, $k \ll n$, and usually $k = 3$ or $4$, meaning it requires negligible computation time. Importantly, the proposed measure is scalable and can be applied to any number of modalities, ranging from 2 to $n$. Interestingly, in the specific case where $k = 2$, the volume computation reduces to the calculation of the area of a parallelogram spanned by two vectors with the origin in the hyperdimensional space. Notably, this area is mathematically linked to the angle between the two vectors, thus establishing a direct correspondence between volume and previous similarity methods in the $k = 2$ case. The extension of this concept to higher-dimensional volumes involving $k > 2$ vectors generalizes cosine and sine similarities to more complex inter-vector relationships. A formal derivation of this generalization is provided in Appendix A.

## 3.4 GRAM-BASED CONTRASTIVE LOSS FUNCTIONS

In this subsection, we completely rethink multimodal representation alignment, by introducing the proposed GRAM-based contrastive loss functions for representation learning and alignment. We define a novel multimodal contrastive loss function completely relying on the aforementioned volume computation. We then describe the additional loss function to enhance the proposed method further.

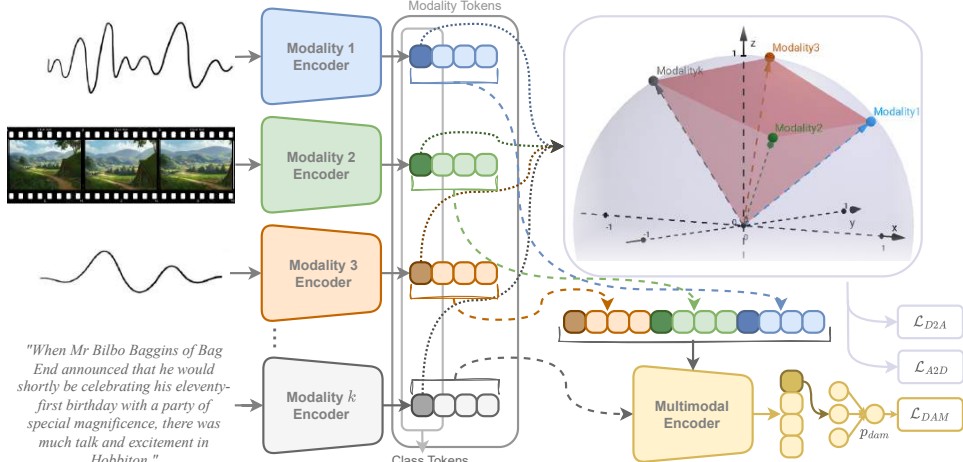

Figure 2: GRAM-based model architecture. Class tokens from each modality are involved in shaping the $k$-dimensional parallelotope, whose volume indicates the semantic alignment of the modalities. All the tokens are then involved in the multimodal encoder to enhance the predictions. The model is pretrained with the proposed Gramian multimodal contrastive losses $\mathcal{L}_{D2A}$ and $\mathcal{L}_{DAM}$.

**GRAM Multimodal Contrastive Loss.** The core of our novel proposal is the volume computation as a measure for the semantic alignment of vectors. Therefore, we exploit the volume as defined in Eq. 4 into the contrastive loss of Radford et al. (2021). Such novel loss encourages volume minimization computed by taking all the modalities into account in a joined fashion, ensuring the semantic alignment of the modalities altogether. After extracting the embedding vectors using encoder functions as defined in Section 3.1, we normalize them. In this way, each modality $M_i$ is represented in a latent embedding vector $\mathbf{m}_i$ and the norm of such vector is always 1. We select an anchor $\mathbf{a}$ choosen as one of the $k$ modalities. Therefore, the proposed GRAM multimodal contrastive loss is:

$$\mathcal{L}_{D2A} = -\frac{1}{B} \sum_{i=1}^{B} \log \frac{\exp(-\mathrm{Vol}(\mathbf{a}_i, \mathbf{m}_{2i}, \dots, \mathbf{m}_{ki})/\tau)}{\sum_{j=1}^{K} \exp(-\mathrm{Vol}(\mathbf{a}_j, \mathbf{m}_{2i}, \dots, \mathbf{m}_{ki})/\tau)}, \tag{5}$$

$$\mathcal{L}_{A2D} = -\frac{1}{B} \sum_{i=1}^{B} \log \frac{\exp(-\mathrm{Vol}(\mathbf{a}_i, \mathbf{m}_{2i}, \dots, \mathbf{m}_{ki})/\tau)}{\sum_{j=1}^{K} \exp(-\mathrm{Vol}(\mathbf{a}_i, \mathbf{m}_{2j}, \dots, \mathbf{m}_{kj})/\tau)}. \tag{6}$$

The volume minimization ensures that the model aligns all the modalities toward a common goal, with the decisive geometric guarantee that the intermodalities converge. By normalizing the vectors to unitary norm, we prevent degenerate cases where the $k$-dimesnional parallelotope collapses toward the origin of the Cartesian coordinate system, and we secure the minimization of the volume by moving vector directions towards each other.

**On the anchor selection:** The choice of anchor modality plays a crucial role in the final evaluation metrics. An anchor can either be a single modality (e.g., text, which is common in tasks involving text-to-data problems) or a multimodal anchor, consisting of two or more modalities combined (e.g., both video and audio). Selecting a specific modality as the anchor is used to direct attention toward it, effectively modeling the entire latent space around that modality.

In Eq 5 and Eq 6, we consider the case of single modality anchor $\mathbf{a}$ choosen as one of the $k$ modalities. Therefore, $\mathbf{a}_x$ refers to the embeddings of the anchor modality of $x$-th sample in the batch, while $\mathbf{m}_{xy}$ refers to the embedding of $x$-th modality of the $j$-th sample in the batch. $B$ is the batch size, $\tau$ learnable scaling parameter.

**Data-Anchor Matching Loss.** In addition to the proposed volume-based multimodal loss function, we employ the supplementary data-anchor matching loss. This loss aims at encouraging the model

to infer whether a pair of anchors and data is matched or not. To compute such matching, any previous encoder model can be involved. Suppose to choose the single-modal text anchor, the text encoder can act as a multimodal encoder, whose input is caption tokens and with the data features as conditioning through cross-attention layers. In this way, we obtain data features by concatenating unpooled features from all the encoders along the sequential dimension. At the bottom of the multimodal encoder, an MLP layer returns binary predictions $p_{dam}$. These predictions are compared with true labels $y$ through binary cross entropy. $y = 1$ when anchor and all other modalities are matched, $y = 0$ otherwise. We follow Li et al. (2021) in the hard negative mining strategy, obtaining the following data-caption matching loss:

$$\mathcal{L}_{DAM} = \mathbb{E}_{(\mathbf{a},\mathbf{m}_2,\cdots,\mathbf{m}_k)\sim(A,M_2,\cdots,M_k)} \left[ y \log p_{dam} + (1-y) \log(1 - p_{dam}) \right]. \tag{7}$$

**GRAM-based Model.** The proposed GRAM Multimodal Model is pretrained with the volume-based contrastive loss functions introduced in Eq. 5. We additionally involve the data-anchor matching loss in Eq. 7 to refine the model predictions. Figure 2 shows the architecture of the model, which is based on VAST Chen et al. (2023c) encoders. For each modality, we map the class tokens into the higher-dimensional latent space and compute the volume as alignment measure. Meanwhile, all the modality tokens of the anchor flow into the multimodal encoder, and the tokens from the other modalities serve as conditioning for the final binary predictions. The final pretraining loss is a combination of $\mathcal{L}_{D2A}$, $\mathcal{L}_{A2D}$, and $\mathcal{L}_{DAM}$ as:

$$\mathcal{L}_{TOT} = \frac{1}{2} \left( \mathcal{L}_{D2A} + \mathcal{L}_{A2D} \right) + \lambda \mathcal{L}_{DAM}, \tag{8}$$

where $\lambda = 0.1$, following Chen et al. (2023c). We provide the details of the models in Appendix B.

### 3.5    GRAM as Model Performance Metric

As GRAM is a measure of the multimodal embedding space alignment, it can also serve as a metric for evaluating large multimodal models. Indeed, the more aligned the multimodal latent space, the better the model will perform in downstream tasks, as it possesses rich semantic information. We prove the validity of these claims by extracting the embeddings from different multimodal models, Language-Bind (LB), VAST, and GRAM-based model, and computing the zero-shot (zs) and fine-tuned (ft) Recall at 1 (R@1). Then, we compute the proposed GRAM, which we rescale for visualization purposes and use $1 - \text{GRAM}$. Figure 3 shows the average, over 1000 test samples, of R@1 (the higher the better) and $1 - \text{GRAM}$ (same) for the video, audio, and text embeddings on MSR-VTT. The linear regression line is computed and added in grey dashed style.

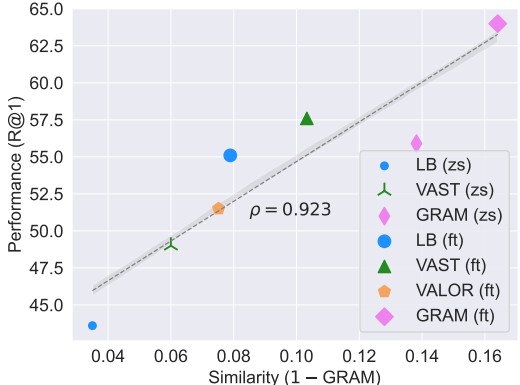

Figure 3: The proposed GRAM similarity is strongly correlated ($\rho = 0.923$) with large multimodal models performance in downstream tasks.

The plot shows a strong correlation between the two metrics, confirmed from the Pearson correlation coefficient ($\rho$) equal to 0.923. This means that the volume of the $k$-dimensional parallelotope spanned by the modality embeddings of every model is strongly linked to the way such a model will perform in the task. Therefore, these results prove that the proposed GRAM is a good metric for large multimodal model performance and that it can be involved as such in future research.

## 4    Experimental Evidences

In this Section, we present the main results of the proposed GRAM contrastive loss and model in downstream tasks. In addition, we show how the multimodal latent space built with GRAM is more meaningful and disentangled with respect to others.

Table 1: **Zero-shot** multimodal text-to-video (T2V) and video-to-text (V2T) retrieval results in terms of Recall at 1 score (R@1). Increment points computed wrt VAST with same modalities.

| | | MSR-VTT | | DiDeMo | | ActivityNet | | VATEX | |
|---|---|---|---|---|---|---|---|---|---|
| | Modality | T2V | V2T | T2V | V2T | T2V | V2T | T2V | V2T |
| UMT Liu et al. (2022) | T-V | 33.3 | - | 34.0 | - | 31.9 | - | - | - |
| OmniVL Wang et al. (2022a) | T-V | 34.6 | - | 33.3 | - | - | - | - | - |
| UMT-L Li et al. (2023) | T-V | 40.7 | 37.1 | 48.6 | 49.9 | 41.9 | 39.4 | - | - |
| TVTSv2 Zeng et al. (2023) | T-V | 38.2 | - | 34.6 - | - | - | - | - | - |
| ViCLIP Wang et al. (2023) | T-V | 42.4 | 41.3 | 18.4 | 27.9 | 15.1 | 24.0 | - | - |
| VideoCoCa Yan et al. (2022) | T-V | 34.3 | 64.7 | - | - | 34.5 | 33.0 | 53.2 | 73.6 |
| Norton Lin et al. (2024) | T-V | 10.7 | | - | - | - | - | - | - |
| ImageBind Girdhar et al. (2023) | T-V | 36.8 | - | - | - | - | - | - | - |
| InternVideo-L Wang et al. (2022b) | T-V | 40.7 | 39.6 | 31.5 | 33.5 | 30.7 | 31.4 | 49.5 | 69.5 |
| HiTeA Ye et al. (2022) | T-V | 34.4 | - | 43.2 | - | - | - | - | - |
| mPLUG-2 Xu et al. (2023) | T-V | 47.1 | - | 45.7 | - | - | - | - | - |
| VideoPrism-b Zhao et al. (2024) | T-V | 51.4 | 50.2 | - | - | 49.6 | 47.9 | 62.5 | 77.1 |
| LanguageBind Zhu et al. (2024) | T-V | 44.8 | 40.9 | 39.9 | 39.8 | 41.0 | 39.1 | - | - |
| VAST Chen et al. (2023c) | T-VA | 49.3 | 43.7 | 49.5 | 48.2 | 51.4 | 46.8 | 80.0 | 77.3 |
| VAST Chen et al. (2023c) | T-VAS | 50.7 | 49.0 | - | - | - | - | 82.1 | 78.7 |
| GRAM Model (Ours) | T-V | 52.8 | 49.5 | 54.0 | **52.3** | 58.9 | **50.9** | 81.1 | 79.0 |
| GRAM Model (Ours) | T-VA | 54.2 (+4.9) | 50.5 (+6.7) | **54.2** (+4.7) | 52.2 (+4) | **59.0** (+7.6) | 50.4 (+3.6) | **83.9** (+3.9) | 79.2 (+1.9) |
| GRAM Model (Ours) | T-VAS | **54.8** (+4.1) | **52.9** (+3.9) | - | - | - | - | 83.5 (+1.4) | **82.7** (+4.0) |

Table 2: **Finetuning** multimodal text-to-video (T2V) and video-to-text (V2T) retrieval results in terms of Recall at 1 score (R@1). Increment points computed wrt VAST with same modalities.

| | | MSR-VTT | | DiDeMo | | ActivityNet | | VATEX | |
|---|---|---|---|---|---|---|---|---|---|
| | Modality | T2V | V2T | T2V | V2T | T2V | V2T | T2V | V2T |
| UMT-L Li et al. (2023) | T-V | 58.8* | 58.6* | 70.4* | 65.7* | 66.8* | 64.4* | 72.0* | 86.0* |
| CLIP4Clip Luo et al. (2021) | T-V | 45.6 | 45.9 | 43.0 | 43.6 | 40.3 | 41.6 | 63.0 | 78.3 |
| ViCLIP Wang et al. (2023) | T-V | 52.5 | 51.8 | 49.4 | 50.2 | 49.8 | 48.1 | - | - |
| InternVideo-L Wang et al. (2022b) | T-V | 55.2* | 57.9* | 57.9* | 59.1* | 62.2* | 62.8* | 71.1* | 87.2* |
| HiTeA Ye et al. (2022) | T-V | 46.8 | - | 56.5 | - | - | - | - | - |
| mPLUG-2 Xu et al. (2023) | T-V | 53.1 | - | 56.4 | - | - | - | - | - |
| VALOR-L Chen et al. (2023b) | T-VAS | 54.4 | - | 57.6 | - | 63.4 | - | 76.9 | - |
| TEFAL Ibrahimi et al. (2023) | T-VA | 52.0 | - | - | - | - | - | 61.0 | - |
| Bimodal T2M Arora et al. (2024) | T-VA | 36.8 | - | - | - | - | - | - | - |
| T-MASS Wang et al. (2024a) | T-VA | 52.7 | - | 53.3 | - | - | - | 65.6 | - |
| vid-TLDR Choi et al. (2024) | T-V | 58.5* | - | 70.4* | - | 65.2* | - | - | - |
| VAST Chen et al. (2023c) | T-VA | 55.8 | 57.6 | 65.6 | 62.0 | 68.8 | 66.7 | 86.9 | 84.1 |
| VAST Chen et al. (2023c) | T-VAS | 56.6 | 57.6 | - | - | - | - | 87.5 | 84.0 |
| GRAM Model (Ours) | T-V | 55.7 | 56.4 | 66.4 | 63.2 | 66.5 | 64.6 | 84.4 | 81.6 |
| GRAM Model (Ours) | T-VA | 58.4 (+2.6) | 59.0 (+1.4) | **67.3** (+1.7) | **63.5** (+1.5) | **69.9** (+1.1) | **66.9** (+0.2) | 87.0 (+0.1) | **84.6** (+0.5) |
| GRAM Model (Ours) | T-VAS | **64.0** (+7.4) | **64.8** (+7.2) | - | - | - | - | **87.7** (+0.2) | 84.2 (+0.2) |

*Finetuning and evaluation with 12 frames.

## 4.1 EXPERIMENTS SETUP

We build the GRAM model with VAST models Chen et al. (2023c) as backbone. Therefore, the text, audio, and video encoders are BERT-B , BEATs Chen et al. (2023a), and EVAClip-ViT-G Sun et al. (2023), respectively, with a total number of parameters equal to 1B. Obviously, we remove VAST fusing layers as GRAM does not require them due to its ability to truly model the multimodal latent space. Starting from VAST pretraining models, we further pretrain those on a small subset of VAST27M Chen et al. (2023c) comprising 150k samples using our defined loss functions. This operation is useful to reshape latent space already built by VAST using our GRAM-based contrastive loss function. We inherit from VAST the learned semantics space and we reshape it using our novel loss function. We set the batch size to 256 and a single epoch pretraining on 4 NVIDIA A100 cards.

As downstream datasets, we consider several well-known multimodal benchmarks that can be divided into three categories: (i) three-modal video-based, such as DiDeMo and ActivityNet, in which the crucial modality is video and the two other modalities (audio and text) are supportive; (ii) four modal video-based, such as MSR-VTT and VATEX, in which video is the main modality but also audio, text, and subtitles are supportive; and (iii) audio-based, like AudioCaps and VGGSound, in which the audio modality is the most relevant, while also video and text contain interesting information. Details about datasets, samples, and resolutions are in Appendix B.

## 4.2 RESULTS AND DISCUSSION

**Multimodal Text-to-Video Retrieval.** Table 1 and Table 2 show the improvements that the proposed GRAM-based contrastive loss brings to large multimodal models in both zero-shot and fine-tuning scenarios, respectively. The proposed GRAM model surpasses by a large margin the wide

Table 3: Zero-shot multimodal text-to-audio retrieval results on AudioCaps and zero-shot audio-text classification results on VGGSound 5K.

| | Modality | AudioCaps | | VGGSound 5K | |
|---|---|---|---|---|---|
| | | R@1 | R@10 | Acc@1 | Acc@10 |
| AVFIC Nagrani et al. (2022) | T-A | 8.7 | 37.7 | - | - |
| AVFIC Nagrani et al. (2022) | T-AV | 10.6 | 45.2 | - | - |
| VIP-ANT Zhao et al. (2022) | T-A | 27.7 | 37.7 | - | - |
| ImageBind Girdhar et al. (2023) | T-A | 9.3 | 42.3 | - | - |
| LanguageBind Zhu et al. (2024) | T-V | - | - | 37.2 | 62.0 |
| LanguageBind Zhu et al. (2024) | T-A | 19.7 | 67.6 | 23.8 | 57.1 |
| VAST Chen et al. (2023c) | T-V | - | - | 38.7 | 72.8 |
| VAST Chen et al. (2023c) | T-A | - | - | 25.6 | 56.2 |
| VAST Chen et al. (2023c) | T-AV | 32.1 | 65.4 | 39.6 | 74.5 |
| GRAM Model (Ours) | T-AV | **33.2** (+1.2) | **75.3** (+9.9) | **40.6** (+1.0) | **78.1** (+3.6) |

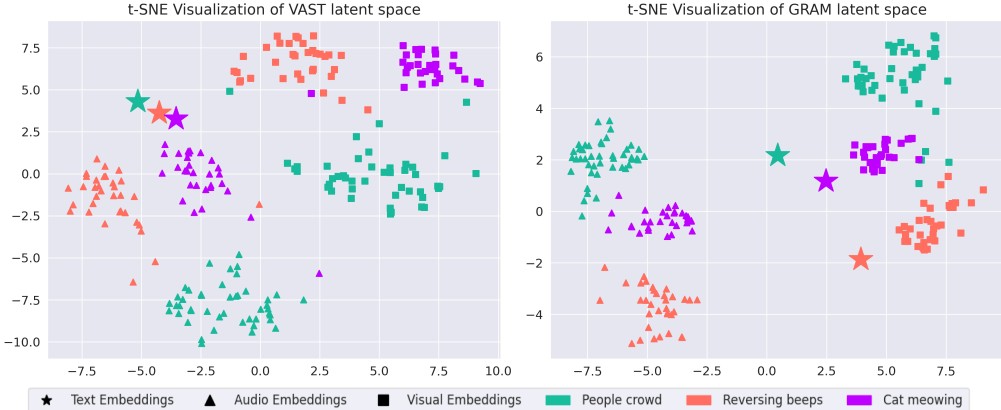

Figure 4: t-SNE visualization on VGGSound of VAST, cosine-based, (left) and GRAM (right) latent spaces. GRAM better models the latent space, resulting in a space more disentangled and highly interpretable. Class clusters are easily recognizable in space, while video (squares) and audio (triangles) modalities that are closer to the classification text (star).

set of comparison methods in every dataset we test and in both cases. In the zero-shot scenario, the GRAM-based model improves the Recall scores with respect to its cosine-based counterpart VAST by an average of 4.5 points, representing a true consistent advancement. These crucial results in zero-shot scenarios mean that the GRAM latent space is much more aligned and semantically separated than the one built with cosine similarity. The meaningfulness of the unified semantic space is clear in finetuning results too, in which the proposed model improves the performance up to 7.4 points of R@1 in the MSR-VTT dataset. The largest part of previous models can only consider pairs of modalities T-V or present fusing strategies to consider triplets T-VA, as they solely rely on the cosine similarity between the two. As a consequence, such models cannot fully exploit the semantic space that multiple modalities define. Instead, the GRAM-based model is pretrained to semantically align all the modalities, shaping a more informative, representative, and meaningful latent space. Finetuning results may be strongly affected by the quality of the original training data. We find that several datasets like DiDeMo and ActivityNet has audio modality semantically very different from the video one probably due to the random selection from YouTube. MSR-VTT, on the other hand, shows higher correlation among multiple modalities that the GRAM model is able to exploit. This outcome demonstrates the importance of modeling multimodal latent spaces in a unified way and the contribution of multiple modalities is pivotal to retrieve the correct data. Additional results and visualizations, including real-world examples in Appendix B, particularly in Fig. 6.

**Multimodal Text-to-Audio Retrieval & Classification.** Table 3 shows the improvements of the GRAM-based model in audio-to-text retrieval (recall at 1 and 10 in AudioCaps) and audio-text classification (accuracy at 1 and 10 in VGGSound). In these scenarios, the proposed GRAM-based model outperforms all the comparison methods. Specifically, GRAM surpasses VAST by up to 9.9 points of R@10, meaning the space that GRAM builds is much more aligned and disentangled than other methods. Table 3 reveals the large adaptability of the GRAM method, which can easily be employed in different datasets and tasks in zero-shot scenarios improving methods performance.

### 4.3 Visualizing the Aligned GRAM Latent Space

To demonstrate the capabilities of the GRAM-based contrastive loss introduced in Section 3.4, we analyze the latent embedding space using t-SNE visualization. For this purpose, we utilize the VGGSound dataset, as it is the only dataset among those considered that includes class labels for its samples, favoring a clearer representation. We visualize the top three classes with the most examples in our downloaded portion. For each class, we extract the embeddings using both VAST and the proposed GRAM-based model for the class labels (text modality), as well as for visual frames and audio samples.

Figure 4 presents the resulting visualizations, where VAST embeddings are on the left, and GRAM space is on the right. The embedding space generated by the GRAM model is significantly more disentangled, with the three modalities of each class well-aligned around the text modality (star symbol). In contrast, the latent space produced by the VAST model appears more scattered, with the different modalities dispersed throughout the embedding space, exhibiting less alignment. Figure 4 provides the visualization of our intuitions, i.e. that GRAM is able to effectively model the multimodal latent space according to the semantics of different modalities. Therefore, in GRAM, all the modalities contribute to the correct classification, resulting in improved performance.

### 4.4 GRAM Scaling to More Modalities

To experimentally confirm the proof that GRAM holds from $2$ to $n$ modalities, we include additional modalities in the MSR-VTT dataset. The largest part of models employ solely the video to correctly retrieve the textual caption. Therefore, first, we test GRAM with these two modalities only, proving that, in this case, the volume computation degenerates to the area of the triangle and that this is still a reliable metric. Following, we add one-by-one the other modalities like audio and subtitles. Table 4 shows that GRAM effectively works for two modalities and that enriching the latent space with more modalities improves the performance from R@1 equal to 52.8 up to

Table 4: GRAM naturally scales to more modalities, shaping a semantically richer latent space that contribute in improving the model performance in downstream tasks. Text-to-Video zero-shot results on MSRVTT

| Text | Video | Audio | Sub. | Depth | R@1 |
|---|---|---|---|---|---|
| ✓ | ✓ | | | | 52.8 |
| ✓ | ✓ | ✓ | | | 54.1 |
| ✓ | ✓ | ✓ | ✓ | | 54.8 |
| ✓ | ✓ | ✓ | ✓ | ✓ | **55.3** |

54.8 in text-to-video retrieval. To further stress the proposed GRAM claims, we insert an additional modality to the four already present in MSR-VTT. We employ ChronoDepth Shao et al. (2024) to extract depth maps from video frames, and add a head to the GRAM vision encoder to process this additional modality. Once pretrained with five modalities on the subset of VAST27M, we perform zero-shot on MSR-VTT with all the modalities. Interestingly, not only the method still perfectly works in defining the 5-dimensional parallelotope spanned by the vectors and in computing the volume, but also further increase the performance up to a recall of 55.3. These results prove two key factors of this work: (i) the intuition that more modalities contribute to a richer semantic space and are often crucial in correctly performing donwstream tasks; (ii) the mathematical proof that the proposed volume computation naturally scales to more modalities beyond the classical 2 or 3 ones.

## 5 Conclusion

In conclusion, we presented GRAM, a fundamentally new measure for multimodal representation learning and alignment that operates in the higher-dimensional space spanned by all modality embeddings. By modeling the alignment through the volume of a parallelotope formed by $k$ modality vectors, GRAM captures richer semantic relationships than traditional pairwise methods like cosine similarity. Furthermore, we introduced a novel GRAM-based contrastive loss, which leverages this geometric alignment to guide multimodal models in shaping a unified embedding space. The model pretrained using our loss outperform state-of-the-art methods by significant margins across multiple tasks, confirming that GRAM generalizes well to a wide range of modalities and tasks. This work represents a significant advancement in multimodal representation learning and alignment by addressing the limitations of pairwise alignment and providing a mathematically grounded, flexible solution for aligning any number of modalities. We believe that GRAM opens up new directions for the field, offering a more powerful and general framework for multimodal understanding and providing new insights into the structure of multimodal spaces.

## REPRODUCIBILITY STATEMENT

In this paper, we present a novel approach to multimodal representation learning. To ensure that our work can be easily reproduced and built upon by the research community, we have taken several key steps. First, the source code implementing our multimodal representation learning model is available as part of the supplementary materials. The code includes all scripts necessary for training, evaluation, and data processing, while pretrained models will be released after reviewing process. Experimental settings and hyperparameters are available in the supplementary materials. We use publicly available datasets and details on training, validation, and testing splits are reported in supplementary materials. In terms of theoretical contributions, we include clear explanations of assumptions and complete proofs for all claims made in the paper. Formal derivations and justifications can be found in the appendix for further verification. Finally, we also provide details about the hardware and software environment used in our experiments.

By providing detailed descriptions, open-source code, and clear theoretical justifications, we aim to make our work on multimodal representation learning fully reproducible and accessible to the broader research community.

## ACKNOWLEDGEMENT

The work of G. Cicchetti, E. Grassucci, and D. Comminiello work was partially supported by the European Union under the Italian National Recovery and Resilience Plan (NRRP) of NextGenerationEU, Mission 4, Component 2, Investment 1.3, partnership on "Telecommunications of the Future" (PE00000001 - program "RESTART") and by the partnership on "Future Artificial Intelligence Research" (PE00000013 – SPOKE 5 - CUP B53C22003980006 - FAIR: High Quality AI). Additionally, the work of L. Sigillo was partly supported by "Ricerca e innovazione nel Lazio - incentivi per i dottorati di innovazione per le imprese e per la PA - L.R. 13/2008" of Regione Lazio, Project "Deep Learning Generativo nel Dominio Ipercomplesso per Applicazioni di Intelligenza Artificiale ad Alta Efficienza Energetica", under grant number 21027NP000000136.

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

# A    THEORETICAL APPENDIX

## A.1    THE GRAMIAM COMPUTES THE VOLUME OF ANY $k$-DIMENSIONAL POLYTOPE

***Theorem 1*: Volume of the $k$-dimensional parallelotope.** Given $\mathbf{v}_1, \ldots, \mathbf{v}_k$ be $k$ vectors in $\mathbb{R}^n$ forming a $k$-dimensional parallelotope. $\mathbf{v}_1, \ldots, \mathbf{v}_k$ can be arranged as column vectors in a matrix $\mathbf{A} = (\mathbf{v}_1, \ldots, \mathbf{v}_k)$ and the Gram matrix $\mathbf{G}$ is computed as in 3. The determinant of the Gram matrix, also called the Gramian, is the square of the volume of the $k$-dimensional parallelotope formed by the vectors Gantmacher (1959):

$$\text{Vol}(\mathbf{v}_1, \ldots, \mathbf{v}_k) = \sqrt{\det \mathbf{G}(\mathbf{v}_1, \ldots, \mathbf{v}_k)}. \tag{9}$$

Note the similarity to the norm of a vector; in fact, $\mathbf{v}(\mathbf{v}) = \|\mathbf{v}\|$, so the Gramian of a single vector is the length, i.e., the 1-dimensional volume, of that vector.

The Gram matrix $\mathbf{A}^\top \mathbf{A}$ captures all the geometric information about the vectors $\mathbf{v}_1, \ldots, \mathbf{v}_k$ : the length of the vectors and the angles between them. We claim that this is enough information to easily determine the volume.

To mathematically prove the theorem we have to distinguish three different cases:

- Consider the case where $k = n$; then

$$\det \mathbf{G}(\mathbf{v}_1, \ldots, \mathbf{v}_k) = \det \mathbf{A}^\top \mathbf{A} = (\det \mathbf{A})(\det \mathbf{A}) = |\mathbf{A}|^2.$$

$$\mathbf{G}(\mathbf{v}_1, \ldots, \mathbf{v}_k) \geq 0$$

Since $\text{Vol}(\mathbf{v}_1, \ldots, \mathbf{v}_k) = |\det \mathbf{A}|$, we obtain:

$$\text{Vol}(\mathbf{v}_1, \ldots, \mathbf{v}_k) = \sqrt{\det \mathbf{A}^\top \mathbf{A}} = \sqrt{\mathbf{G}(\mathbf{v}_1, \ldots, \mathbf{v}_k)}$$

- Consider the case where $k < n$; then the above holds by simply restricting to a $k$-dimensional subspace containing them; Given $\mathbf{v}_1, \ldots, \mathbf{v}_k$, where $k \geq n$, assume that they are linearly independent and pick an orthonormal basis $\mathbf{w}_1, \ldots, \mathbf{w}_k$ for their span, $\mathbf{W}$. Extend to an orthonormal basis $\mathbf{w}_1, \ldots, \mathbf{w}_k$ for $\mathbb{R}^n$; the matrix $\mathbf{O}$ sending $\mathbf{w}_i \longmapsto \mathbf{e}_i$ is an orthogonal change of coordinates, so it does not change inner products: $\mathbf{O}(\mathbf{v}) \cdot \mathbf{O}(\mathbf{w}) = \mathbf{v} \cdot \mathbf{w}$. Let $\mathbf{v}_i' = \mathbf{O}(\mathbf{v}_i)$ then

$$(\mathbf{v}_i') = \begin{bmatrix} m_{1i} \\ m_{2i} \\ \vdots \\ m_{ki} \\ 0 \\ \vdots \\ 0 \end{bmatrix}$$

Since $v_i$ is in the span of $\mathbf{w}_1, \ldots, \mathbf{w}_k$. The set $\mathbf{v}_1', \ldots, \mathbf{v}_k'$ visibly lies in the k-dimensional subspace of vectors with last $n - k$ coordinates zero. Effectively we are restricting to the subspace $\mathbf{W}$. Since O is orthogonal,

$$\text{Vol}(\mathbf{v}_1', \ldots, \mathbf{v}_k') = \text{Vol}(\mathbf{v}_1, \ldots, \mathbf{v}_k)$$

and $\mathbf{v}_i' \cdot \mathbf{v}_j' = \mathbf{v}_i \cdot \mathbf{v}_j$, so

$$\sqrt{\mathbf{G}(\mathbf{v}_1, \ldots, \mathbf{v}_k)} = \text{Vol}(\mathbf{v}_1', \ldots, \mathbf{v}_k') = \text{Vol}(\mathbf{v}_1, \ldots, \mathbf{v}_k)$$

as desired.

- When $k > n$, the vectors are linearly dependent, so the $k$-volume of the parallelotope is zero, since the determinant of $\mathbf{A}$ matrix is zero.

## A.2 Theoretical Advantages of Volume computation wrt Cosine Similarity

Let us consider the simple case with three modalities: Text ($T$), Video ($V$), and Audio ($A$). The Gram Matrix is equal to:

$$\mathbf{G} = \begin{bmatrix} TT & TA & TV \\ AT & AA & AV \\ VT & VA & VV \end{bmatrix}$$

Now, let us compute the determinant of the matrix $\mathbf{G}$:

$$\det(G) = TT\cdot(AA\cdot VV - AV\cdot VA) - TA\cdot(AT\cdot VV - AV\cdot VT) + TV\cdot(AT\cdot VA - AA\cdot VT) \tag{10}$$

Recall that $T, V, A$ embeddings are normalized to unit norm and so $TT = VV = AA = 1$:

$$\det(\mathbf{G}) = 1\cdot(1 - VA^2) - TA\cdot(AT - AV\cdot VT) + TV\cdot(AT\cdot VA - VT) \tag{11}$$
$$= 1 - VA^2 - TA^2 + TA\cdot AV\cdot VT + TV\cdot AT\cdot VA - TV^2 \tag{12}$$
$$= 1 - VA^2 - TA^2 - TV^2 + 2\cdot TA\cdot AV\cdot VT \tag{13}$$

Therefore, the volume computation through the Gram matrix includes in its computation all the cross-products, resulting in an alignment of all the modalities together. In contrast, current state-of-the-art methods based on cosine similarity only compute the similarities between the modalities and the anchor ($TA$ and $TV$), omitting the similarities between non-anchor modalities ($AV$), which in practice may not be aligned.

## A.3 The volume is related to the angle between the two vectors in the two-modal case

Consider two vectors $\mathbf{v_1}, \mathbf{v_2} \in \mathbb{R}^n$ with $\|\mathbf{v_1}\| = \|\mathbf{v_2}\| = 1$. The Gram matrix $\mathbf{G}$ is given by:

$$\mathbf{G} = \begin{bmatrix} \langle\mathbf{v}_1^\top\mathbf{v}_1\rangle & \langle\mathbf{v}_1^\top\mathbf{v}_2\rangle \\ \langle\mathbf{v}_2^\top\mathbf{v}_1\rangle & \langle\mathbf{v}_2^\top\mathbf{v}_2\rangle \end{bmatrix} \tag{14}$$

Then, compute the determinant of the Gram matrix:

$$\det(\mathbf{G}) = \langle\mathbf{v}_1^\top\mathbf{v}_1\rangle\langle\mathbf{v}_2^\top\mathbf{v}_2\rangle - \langle\mathbf{v}_1^\top\mathbf{v}_2\rangle^2 \tag{15}$$

The volume of the $k$-dimensional parallelotope spanned by the modalities $\mathbf{v}_1, \mathbf{v}_2$ is:

$$\text{Vol} = \sqrt{\det(\mathbf{G})} = \sqrt{\langle\mathbf{v}_1^\top\mathbf{v}_1\rangle\langle\mathbf{v}_2^\top\mathbf{v}_2\rangle - \langle\mathbf{v}_1^\top\mathbf{v}_2\rangle^2} \tag{16}$$

Given that $\mathbf{v}_1$ and $\mathbf{v}_2$ have norm equal to 1, equation 16 reduces to:

$$\text{Vol} = \sqrt{\det(\mathbf{G})} = \sqrt{1 - \langle\mathbf{v}_1^\top\mathbf{v}_2\rangle^2} \tag{17}$$

The cosine similarity is defined as $\cos(\theta) = \langle\mathbf{v}_1^\top\mathbf{v}_2\rangle/\|\mathbf{v}_1\|\cdot\|\mathbf{v}_2\|$, however, considering the unitary norm of the embeddings vectors, the cosine similarity reduced to $\cos(\theta) = \langle\mathbf{v}_1^\top\mathbf{v}_2\rangle$. Therefore, equation 17 becomes:

$$\text{Vol} = \sqrt{\det(\mathbf{G})} = \sqrt{1 - \cos^2(\theta)} = \sqrt{\sin^2(\theta)} = \sin(\theta). \tag{18}$$

Therefore, in the case of two modalities, the volume computation reduces to the sine of the angle between the two modality vectors.

Table 5: Dataset statistics and hyperparameters. Modalities stand for T: text, V: video, A: audio, S: subtitles, D: depth. # Frames refers both to training and inference.

| Modality | Benchmark | #Video / #Audio | | | # Frames | # Epochs |
|---|---|---|---|---|---|---|
| | | Train | Val | Test | | |
| Threemodal (T-V-A) | AudioCaps | - | - | 700 | 8 | - |
| | VGGSound | - | - | 5000 | 8 | - |
| | DiDeMo | 8394 | 1065 | 1003 | 8 | 40 |
| | ActivityNet | 10009 | - | 4917 | 8 | 20 |
| Fourmodal (T-V-A-S) | MSR-VTT | 9000 | - | 1000 | 8 | 4 |
| | VATEX | 14060 | - | 431 | 8 | 3 |
| Fivemodal (T-V-A-S-D) | MSR-VTT | 9000 | - | 1000 | 8 | 4 |

# B  EXPERIMENTAL APPENDIX

## B.1  EXPERIMENTAL DETAILS

We utilize several benchmark datasets for our downstream tasks:

**MSR-VTT** Xu et al. (2016) comprises 10000 video clips accompanied by 200000 captions, encompassing a diverse range of subjects including human activities, sports, and natural landscapes.

**VATEX** Wang et al. (2019) consists of 41250 video clips derived from the Kinetics-600 dataset Kay et al. (2017) and 825000 sentence-level descriptions. Since a large part of the dataset is now unavailable online due to removed or private videos thus we use only a portion of the original dataset composed of 14491 samples.

**DiDeMo** Hendricks et al. (2017) includes 10000 long-form videos from Flickr, with each video annotated by four temporally ordered short sentences. The dataset is uniquely annotated with natural language descriptions corresponding to distinct moments within each video. For each video, four short sentences are provided, arranged in temporal order to describe specific events or scenes.

**ActivityNet** Caba Heilbron et al. (2015) comprises 20000 long-form videos (with an average duration of 180 seconds) from YouTube and 100000 captions. It has a diverse range of 200 human activity classes, spanning daily activities, sports, and various other complex human behaviors. ActivityNet is annotated with both activity labels and temporal boundaries, providing fine-grained information about when specific activities occur within each video.

**AudioCaps** Kim et al. (2019) is a dataset comprising 51000 audio clips, each with a duration of 10 seconds. The dataset annotation structure varies between its subsets: the training set contains one caption per clip, while the validation and test sets are annotated with five captions per clip. We follow the split protocol established by Oncescu et al. (2021) for the text-to-audio retrieval task.

**VGGSound** Chen et al. (2020) is a large-scale audio-visual dataset containing more than 200000 videos sourced from YouTube, encompassing a diverse range of 309 audio classes. Each video clip in the dataset has a duration of 10 seconds and is annotated with a single label corresponding to the predominant sound event occurring within the clip. The dataset covers a wide spectrum of audio events, including human actions, animal vocalizations, natural phenomena, and mechanical sounds. Due to several downloading problems we use only a portion of the original dataset composed of 5000 samples in testing.

For every dataset, we utilize the official split for retrieval tasks, the dataset splits, and the number of frames for fine-tuning and evaluations on all the datasets in Tab. 5. We pretrain the GRAM-based model on a subset of the VAST27M Chen et al. (2023c) dataset comprising 150k random samples with a learning rate of $1e-4$ using the AdamW optimizer with weight decay and batch size of 256. For finetuning we reduce the batch size to 64 and change the number of epochs according to the specific dataset, the complete details are shown in 5.

Table 6: Ablation study on loss functions. Recall at 1 is shown for both Text-to-Video (T2V) and Video-to-Text (V2T) tasks training from scratch on MSR-VTT and ActivityNet datasets.

| | | | | | MSR-VTT | | ActivityNet | |
|---|---|---|---|---|---|---|---|---|
| TV | TA | D2A | A2D | DAM | T2V | V2T | T2V | V2T |
| ✓ | ✓ | ✗ | ✗ | ✗ | 36.4 | 36.7 | 23.6 | 21.3 |
| ✗ | ✗ | ✗ | ✓ | ✓ | 20.9 | 29.0 | 16.3 | 18.9 |
| ✗ | ✗ | ✓ | ✗ | ✓ | 37.1 | 38.7 | 23.6 | 24.5 |
| ✗ | ✗ | ✓ | ✓ | ✗ | 38.0 | 41.2 | 30.0 | 30.0 |
| ✗ | ✗ | ✓ | ✓ | ✓ | **38.9** | **41.9** | **30.2** | **30.1** |

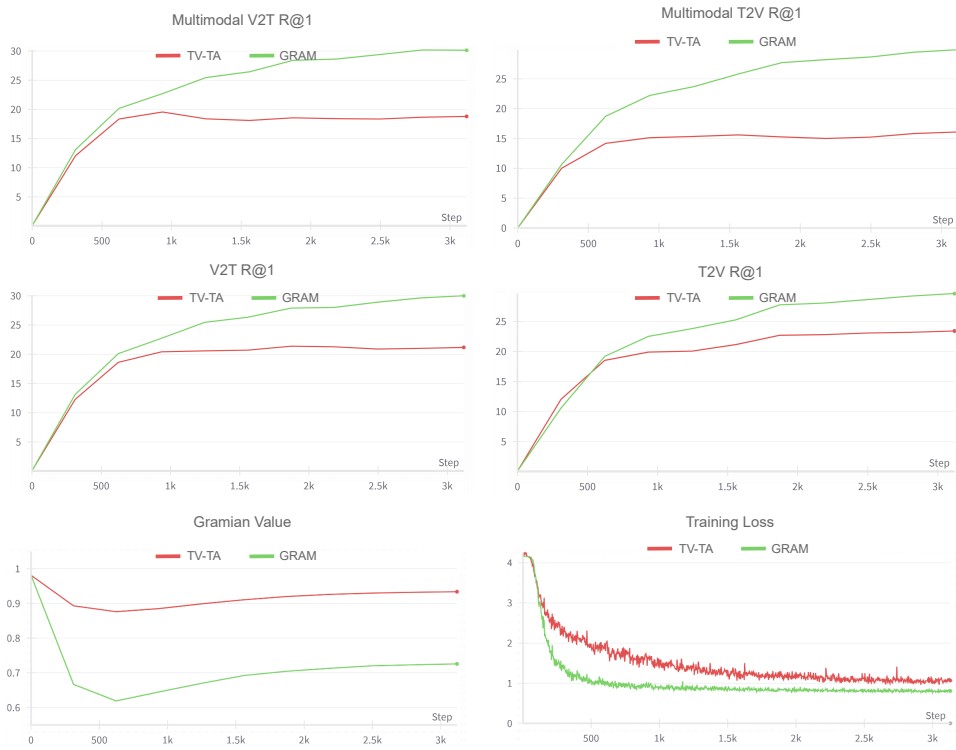

Figure 5: Multimodal V2T/T2V, V2T/T2V, Gramian Value and training loss of GRAM trained with our loss functions and GRAM trained with only TV-TA loss functions. ActivityNet Dataset, training from scratch.

## B.2 ABLATION STUDIES

We perform ablation studies to further validate our experimental results. To evaluate the effectiveness of the introduced volume-based loss functions and the contribute of each one we perform comparative test on MSR-VTT and ActivityNet datasets. Results shown in Tab. 6 are obtained training from scratch our model with different training settings. We train the model for 4 epochs on MSR-VTT and 20 epochs on ActivityNet. The use of our proposed loss funtions allow the model to obtain superior performance with respect to the classical approach in which we use cosine-based contrastive loss functions among pairs of modalities.

To further investigate the superiority of our approach compared to the classical method, which relies on measuring similarity among pairs of modalities, we present several key metrics that provide insights into the behavior of GRAM during training. The gramian value is computed averaging

Table 7: Modality Gap in GRAM. Results computed on extracted embeddings of MSR-VTT.

| Method | VV | TT | AA | VT | TA | VA |
|---|---|---|---|---|---|---|
| VAST (zs) | 0.36 | 0.16 | 0.57 | 0.65 | 0.84 | 0.96 |
| GRAM (zs) | 0.39 | 0.19 | 0.85 | 0.54 | 0.98 | 1.17 |
| VAST (ft) | 0.08 | 0.07 | 0.69 | 0.37 | 0.86 | 0.84 |
| GRAM (ft) | 0.21 | 0.23 | 0.84 | 0.40 | 0.96 | 1.09 |

the volume measure among all true predictions made by GRAM. In Fig. 5, we show the results of training from scratch on the ActivityNet dataset. First, we report the use of only the TV-TA contrastive loss functions (corresponding to row 1 of Tab. 6, represented by the red lines). Next, we present the results when our proposed loss functions are applied (corresponding to row 5 of Tab. 6). As shown, the introduction of GRAM-based loss functions significantly improves the alignment of the entire latent space. This approach not only enhances the alignment between text and video modalities but also outperforms the specialized TV loss. Furthermore, the training loss is much more stable, and convergence occurs earlier, all with the same set of hyperparameters.

## B.3 DISCUSSION ON THE MODALITY GAP

We further explore the phenomenon of the modality gap, as introduced by Liang et al. (2022). In Tab. 7, we report the mean cosine similarity between embeddings of the same modality (VV $\rightarrow$ Vision, TT $\rightarrow$ Text, AA $\rightarrow$ Audio) and the distances between the centroids of different modalities (e.g., VT $\rightarrow$ distance between the Vision centroid and the Text centroid, and so on). The centroids are computed following the methodology outlined in Liang et al. (2022). Consistent with the conclusions of Liang et al. (2022), we observe that the modality gap exists, and its relationship with final performance metrics is complex and not easily interpretable. An empirical hypothesis we propose is that the loss functions we introduce favor two behaviors: i) it squeezes the clusters of each modality more than VAST, indeed the average cosine similarities inside the modalities are higher. ii) It increases the gap among the modalities, probably producing a more sparse latent space. This is clear from the last three columns of Tab. 7, where the distances between modality cluster centroids are shown. Again, as clearly stated in Liang et al. (2022), although GRAM obtains better performance in downstream tasks, there are no mathematical proofs that link such performance to the larger modality gap.

## B.4 ADDITIONAL RESULTS

Figure 6 shows a visualization that demonstrates the effectiveness of the proposed method. Specifically, using four triplets from YouTube consisting of video, audio, and text, we first compute the cosine similarity between text-video and text-audio pairs across all text labels and videos/audios. These results are arranged into a $4 \times 4$ matrix, and a temperature-scaled softmax is applied to the rows to obtain a probability distribution for each text-video (audio) pair assigned by the model.

Next, we apply our proposed GRAM approach to the same dataset, once again performing temperature-scaled softmax on the matrix rows. In this case, the volume matrix is negated, as a lower volume indicates a stronger correspondence among text, video, and audio.

As Fig 6 highlights, conventional cosine-based methods fail in jointly exploiting both audio and video modalities, often misleading when a single modality does not contain enough information for the correct classification. Conversely, the GRAM-based model exploits the semantically aligned multimodal space and jointly leverages all the modalities, leading to a better and correct retrieval in every case.

Additionally, we report Recall@1 and Recall@10 for text-to-video and video-to-text both zero-shot and finetuned experiments in Table 8, Table 9, Table 10, and Table 11. Our GRAM model outperforms every method in the very large set of comparison methods in T2V and V2T both in zero-shot and fine-tuning scenarios. More interestingly, the proposed method always achieves very

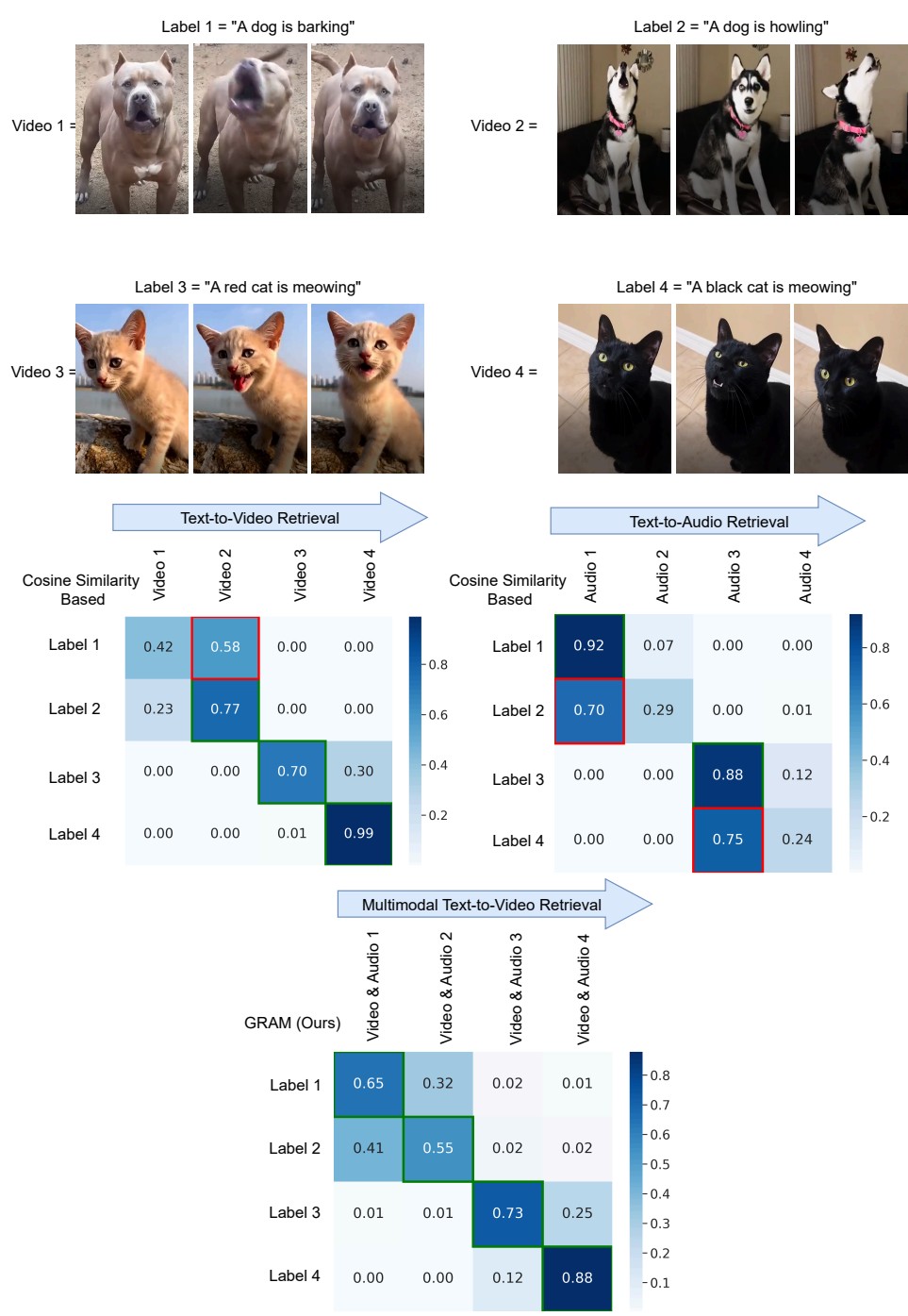

Figure 6: Confusion matrices of cosine-based approach and our proposed GRAM.

high scores of R@10, especially the 100% in Table 10 and Table 11 for the VATEX dataset, meaning that the space the model shapes is extremely aligned.

Table 8: Zero-shot multimodal text-to-video retrieval results. Recall at 1 and Recall at 10.

| Zero-Shot T2V Retrieval | Modality | MSR-VTT | | DiDeMo | | ActivityNet | | VATEX | |
|---|---|---|---|---|---|---|---|---|---|
| | | R@1 | R@10 | R@1 | R@10 | R@1 | R@10 | R@1 | R@10 |
| UMT Liu et al. (2022) | T-V | 33.3 | 66.7 | 34.0 | 68.7 | 31.9 | 72.0 | - | - |
| OmniVL Wang et al. (2022a) | T-V | 42.0 | 73.0 | 40.6 | 74.3 | - | - | - | - |
| UMT-L Li et al. (2023) | T-V | 40.7 | 71.8 | 48.6 | 79.0 | 41.9 | - | - | - |
| TVTSv2 Zeng et al. (2023) | T-V | 38.2 | 73.2 | 34.6 | 71.5 | - | - | - | - |
| ViCLIP Wang et al. (2023) | T-V | 42.4 | - | 18.4 | - | 15.1 | - | - | - |
| VideoCoCa Yan et al. (2022) | T-V | 34.3 | 67.0 | - | - | 34.5 | 76.6 | 53.2 | - |
| Norton Lin et al. (2024) | T-V | 10.7 | 31.6 | - | - | - | - | - | - |
| ImageBind Girdhar et al. (2023) | T-V | 36.8 | 70.0 | - | - | - | - | - | - |
| InternVideo-L Wang et al. (2022b) | T-V | 40.7 | - | 31.5 | - | 30.7 | - | 49.5 | - |
| HiTeA Ye et al. (2022) | T-V | 34.4 | 69.9 | 43.2 | 79.0 | - | - | - | - |
| mPLUG-2 Xu et al. (2023) | T-V | 47.1 | 79.0 | 45.7 | 71.1 | - | - | - | - |
| VideoPrism-b Zhao et al. (2024) | T-V | 51.4 | - | - | - | 49.6 | - | 62.5 | - |
| LanguageBind Zhu et al. (2024) | T-V | 44.8 | 78.7 | 39.9 | 74.6 | 41.0 | 80 | - | - |
| VAST Chen et al. (2023c) | T-VA | 49.3 | 80.0 | 49.5 | 76.9 | 51.4 | 83.6 | 80.0 | 95.7 |
| VAST Chen et al. (2023c) | T-VAS | 50.7 | 74.4 | - | - | - | - | 82.1 | 96.8 |
| GRAM Model (Ours) | T-V | 52.8 | 82.9 | 54.0 | **80.7** | 58.9 | **91.2** | 81.1 | **99.5** |
| GRAM Model (Ours) | T-VA | 54.2 | **83.9** | **54.2** | 79.3 | **59.0** | 91.1 | **83.9** | 98.6 |
| GRAM Model (Ours) | T-VAS | **54.8** | 82.9 | - | - | - | - | 83.5 | 98.8 |

Table 9: Zero-shot multimodal video-to-text retrieval results. Recall at 1 and Recall at 10.

| Zero-Shot V2T Retrieval | Modality | MSR-VTT | | DiDeMo | | ActivityNet | | VATEX | |
|---|---|---|---|---|---|---|---|---|---|
| | | R@1 | R@10 | R@1 | R@10 | R@1 | R@10 | R@1 | R@10 |
| UMT Liu et al. (2022) | T-V | 33.3 | 66.7 | 34.0 | 68.7 | 31.9 | 72.0 | - | - |
| OmniVL Wang et al. (2022a) | T-V | 34.6 | 66.6 | 33.3 | 68.5 | - | - | - | - |
| UMT-L Li et al. (2023) | T-V | 40.7 | - | 24.9 | - | 39.4 | - | - | - |
| TVTSv2 Zeng et al. (2023) | T-V | 38.2 | 73.2 | 34.6 | 71.5 | - | - | - | - |
| ViCLIP Wang et al. (2023) | T-V | 41.3 | - | 27.9 | - | 24.0 | - | - | - |
| VideoCoCa Yan et al. (2022) | T-V | 34.3 | 67.0 | - | - | 34.5 | 76.6 | - | - |
| ImageBind Girdhar et al. (2023) | T-V | 36.8 | 70.0 | - | - | - | - | - | - |
| InternVideo-L Wang et al. (2022b) | T-V | 39.6 | - | 33.5 | - | 31.4 | - | 69.5 | - |
| HiTeA Ye et al. (2022) | T-V | 46.8 | - | 56.5 | - | - | - | - | - |
| VideoPrism-b Zhao et al. (2024) | T-V | 50.2 | - | - | - | 47.9 | - | 76.2 | - |
| LanguageBind Zhu et al. (2024) | T-V | 40.9 | 75.7 | 39.8 | 76.2 | 39.1 | 81.1 | - | - |
| VAST Chen et al. (2023c) | T-VA | 43.7 | 77.1 | 48.2 | 78.6 | 46.8 | 77.4 | 77.1 | 95.2 |
| VAST Chen et al. (2023c) | T-VAS | 49.0 | 76.2 | - | - | - | - | 78.7 | 97.7 |
| GRAM Model (Ours) | T-V | 49.5 | 81.7 | **52.3** | **80.3** | 50.9 | 85.4 | 79.0 | 98.3 |
| GRAM Model (Ours) | T-VA | 50.5 | 82.2 | 52.2 | 78.9 | 50.4 | **85.8** | 79.2 | **99.0** |
| GRAM Model (Ours) | T-VAS | **52.9** | **82.9** | - | - | - | - | **82.7** | 98.1 |

Table 10: Finetuning multimodal text-to-video retrieval results.

| Finetuning T2V Retrieval | | MSR-VTT | | DiDeMo | | ActivityNet | | VATEX | |
|---|---|---|---|---|---|---|---|---|---|
| | Modality | R@1 | R@10 | R@1 | R@10 | R@1 | R@10 | R@1 | R@10 |
| OmniVL Wang et al. (2022a) | T-V | 47.8 | 83.8 | 52.4 | 85.4 | - | - | - | - |
| UMT-L Li et al. (2023) | T-V | 58.8* | 87.1* | 70.4* | 93.5* | 66.8* | 94.9* | 72.0* | - |
| ViCLIP Wang et al. (2023) | T-V | 52.5 | - | 49.4 | - | 49.8 | - | | |
| CLIP4Clip Luo et al. (2021) | T-V | 45.6 | 81.6 | 43.0 | 80.6 | 40.3 | - | 63.0 | - |
| Norton Lin et al. (2024) | T-V | 31.2 | 66.8 | - | - | - | - | - | - |
| InternVideo-L Wang et al. (2022b) | T-V | 55.2* | - | 57.9* | - | 62.2* | - | 71.1* | - |
| HiTeA Ye et al. (2022) | T-V | 46.8 | 81.9 | 56.5 | 89.7 | - | - | - | - |
| mPLUG-2 Xu et al. (2023) | T-V | 53.1 | 84.7 | 56.4 | 85.2 | - | - | - | - |
| TEFAL Ibrahimi et al. (2023) | T-VA | 52.0 | 86.1 | - | - | - | - | 61.0 | 95.3 |
| Bimodal T2M Arora et al. (2024) | T-VA | 36.8 | - | - | - | - | - | - | - |
| T-MASS Wang et al. (2024a) | T-VA | 52.7 | 85.6 | 53.3 | 87.7 | - | - | 65.6 | 97.2 |
| vid-TLDR Choi et al. (2024) | T-V | 58.5* | 86.9* | 70.4* | 94.0* | 65.2* | 94.5* | - | - |
| VAST Chen et al. (2023c) | T-VA | 55.8 | 85.9 | 65.6 | 88.1 | 68.8 | 95.5 | 86.9 | 99.1 |
| VAST Chen et al. (2023c) | T-VAS | 56.6 | 79.4 | - | - | - | - | 87.5 | 99.5 |
| GRAM Model (Ours) | T-V | 55.7 | 86.4 | 66.4 | 89.9 | 66.5 | 96.0 | 84.4 | 99.8 |
| GRAM Model (Ours) | T-VA | 58.4 | 87.0 | **67.3** | **90.1** | **69.9** | **96.1** | 87.0 | 99.5 |
| GRAM Model (Ours) | T-VAS | **64.0** | **89.3** | - | - | - | - | **87.7** | **100.0** |

*Finetuning and evaluation with 12 frames.

Table 11: Finetuning multimodal video-to-text retrieval results.

| Finetuning V2T Retrieval | | MSR-VTT | | DiDeMo | | ActivityNet | | VATEX | |
|---|---|---|---|---|---|---|---|---|---|
| | Modality | R@1 | R@10 | R@1 | R@10 | R@1 | R@10 | R@1 | R@10 |
| UMT-L Li et al. (2023) | T-V | 58.6* | - | 65.7* | - | 64.6* | - | 86.0* | - |
| CLIP4Clip Luo et al. (2021) | T-V | 45.9 | - | 43.6 | - | 41.6 | - | 78.3 | - |
| ViCLIP Wang et al. (2023) | T-V | 51.8 | - | 50.2 | - | 48.1 | - | - | - |
| VAST Chen et al. (2023c) | T-VA | 57.6 | 87.4 | 62.0 | 87.7 | 66.7 | 95.3 | 84.1 | 99.8 |
| VAST Chen et al. (2023c) | T-VAS | 57.6 | 80.2 | - | - | - | - | 84.0 | 99.7 |
| GRAM Model (Ours) | T-V | 56.4 | 87.6 | 63.2 | **91.6** | 64.6 | 95.1 | 81.6 | 99.8 |
| GRAM Model (Ours) | T-VA | 59.0 | 89.1 | **63.5** | 91.4 | **66.9** | **95.4** | **84.6** | **100.0** |
| GRAM Model (Ours) | T-VAS | **64.8** | **91.5** | - | - | - | - | 84.2 | 99.8 |

*Finetuning and evaluation with 12 frames.

