# OpenReview forum: "Gramian Multimodal Representation Learning and Alignment"
_ICLR.cc/2025/Conference — ICLR 2025 Poster_

### Official Review · Reviewer_iJhF · 2024-10-31

**Soundness:** 3
**Presentation:** 2
**Contribution:** 3
**Rating:** 6
**Confidence:** 4

**Summary:**

This paper focuses on the problem where cosine similarity effectively represents the correlation between paired modalities in multimodal learning scenarios, but performs poorly with more than two modalities. It proposes a Gramian Representation Alignment Measure (GRAM) to quantify modality correlation by calculating the volume of a parallelotope. And then the traditional contrastive loss is improved with GRAM, which can hold for 2 to n modality and providing more meaningful alignment with respect to previous similarity measures. Extensive experiments validate the effectiveness of the method.

**Strengths:**

1. This paper proposes a new volume-based modality alignment method, which sounds reasonable and overcomes the limitation of traditional contrastive learning that only computes between paired modalities.
2. Comprehensive evaluation demonstrates the effectiveness of the proposed methods

**Weaknesses:**

1. Lack of the details about how to decide anchor modality for Eq. 5-7.
2. When one modality is selected as anchor, minimizing the cosine similarity between the anchor and all other modalities on hypersphere can also minimize the volume of parallelotope. What is the difference between the two in terms of physical meaning?
3. [A] demonstrated the modality gap from contrastive learning and pointed that varying the modality gap distance has a significant impact in improving the model's downstream zero-shot classification performance and fairness. How to explain this phenomenon by your measure? and is there modality gap from GRAM-based contrastive loss?
4. The setup of the comparison experiment is not entirely fair. Since the proposed model is pretrained with the new loss and data set, the same should be done for the comparison methods. Meanwhile, ablation experiments for different losses ($L_{D2A}$ and $L_{DAM}$) are missing.



[A] Mind the Gap: Understanding the Modality Gap in Multi-modal Contrastive Representation Learning. NeuIPS 2022.

**Questions:**

see in Weaknesses

---

> ### Author Response · Authors · 2024-11-19
> **Response to Points 1 & 2 & 3**
>
> We would like to thank the Reviewer for the insightful comments and the interesting suggestion of studying the modality gap in our paper.
>
> **On the anchor selection.** According to the Reviewer’s suggestion, we added details in subsection 3.4 by inserting a paragraph on the anchor selection. In our experiments, we select as an anchor the text, which has been proven to be the most representative and aligned modality with other modalities [1]. Furthermore, selecting a specific modality as the anchor is used to direct attention toward it, effectively modeling the entire latent space around that modality. Nevertheless and contrary to cosine-based methods, going beyond cosine similarity, GRAM also allows the selection of diverse or multiple modalities.
>
> [1] Bin Zhu, et.al., “LanguageBind: Extending Video-Language Pretraining to N-modality by Language-based Semantic Alignment”, ICLR 2023.
>
> **On the difference between volume and cosine similarity.** As the Reviewer correctly pointed out, in some cases, minimizing the cosine similarity between the anchor and other modalities can also minimize the volume. However, when using the cosine similarity, the modalities are aligned to the anchor, independently from other modalities in a discordant setting. By using the volume, all the modalities are aligned together in a united and simultaneous way. This harmony leads to a more aligned latent space with respect to the cosine similarity, as shown in Figure 4.
>
> This is clear from the Gramian Value plot in Figure 5 Appendix B.2, where the same models are trained with conventional cosine similarity TV-TA losses and with the proposed GRAM. Cosine similarity losses poorly reduce the Gramian value, which indeed remains equal to 0.95 more or less, meaning that while the TV-TA model tries to align the modalities to the textual anchor, they remain quite far from each other, therefore resulting in a high volume.
>
> **On the modality gap.** We would like to thank the Reviewer for the interesting point raised and for allowing us to deeply analyze the latent space produced by GRAM. We compute the modality gap of VAST and GRAM and report the results in Appendix B.3 Table 7.
> We report the mean cosine similarity between embeddings of the same modality (VV → Vision, TT → Text, AA → Audio) and the distances between the centroids of different modalities (e.g., VT → distance between the Vision centroid and the Text centroid, and so on). The centroids are computed following the methodology outlined in [A].
> We also report the table here.
>
> | Method  | VV   | TT   | AA   | VT   | TA   | VA   |
> |---------|------|------|------|------|------|------|
> | VAST (zs) | 0.36 | 0.16 | 0.57 | 0.65 | 0.84 | 0.96 |
> | GRAM (zs) | 0.39 | 0.19 | 0.85 | 0.54 | 0.98 | 1.17 |
> |         |      |      |      |      |      |      |
> | VAST (ft) | 0.08 | 0.07 | 0.69 | 0.37 | 0.86 | 0.84 |
> | GRAM (ft) | 0.21 | 0.23 | 0.84 | 0.40 | 0.96 | 1.09 |
>
> As shown in the Table, we discovered that the modality gap is also present in the latent space generated by GRAM. In line with the conclusions of [A], we observe that the modality gap exists, and its relationship with final performance metrics is complex and not easily interpretable. An empirical hypothesis we propose is that the loss functions we introduce favor two behaviors:
>
> 1. it squeezes the clusters of each modality more than VAST, indeed the average cosine similarities inside the modalities are higher.
>
> 2. It increases the gap among the modalities, probably producing a more sparse latent space.
>
> This is clear from the last three columns of the table, where the distances between modality cluster centroids are shown. Again, as clearly stated in [A], although GRAM obtains better performance in downstream tasks, there are no mathematical proofs that link such performance to the larger modality gap.
>
> Overall, we found the modality gap a very interesting point we were not aware of, and we will definitely investigate it further in future work.

---

> ### Author Response · Authors · 2024-11-19
> **Response to Point 4**
>
> We thank the reviewer for giving us the opportunity to explain this crucial aspect of the work more in detail.
>
> **On the comparison experiments.** As we pointed out in the introduction section of the paper, “the current trajectory of multimodal learning is reaching a plateau, with the sole possible research direction of scaling up models and datasets bringing negligible improvements.”
> Data is a crucial aspect for the entire learning process since better-aligned multimodal input data results in better and more effective learning from the model. The same holds for the number of parameters of the model.
> In fact, most of the new papers propose a model and a new pretraining dataset on which they pretrain the model [1,2,3,4,5]. It is common practice in the literature that, once pretrained, the models are tested in downstream datasets, making comparisons with different models that have been pretrained on different datasets too. Moreover, these pretraining datasets are often huge (27 Milions of videos for VAST, 10 Milions of videos for Language Bind and so on and so forth)  and not totally available due to the updated YouTube policies on video downloads. Nevertheless, our aim is not to develop a novel model or pretrain it on a huge and new dataset but rather to propose a novel perspective and a novel measure for multimodal learning that brings theoretical guarantees and consistently improves performance in any downstream task.
>
> Having said that, we understand the doubts raised by the Reviewer and, for this reason, we further investigate the contribution of our loss function in a controlled environment. We take the VAST model (chosen as an illustrative example of multimodal models trained with cosine-based loss functions) and we train it from scratch on MSR-VTT and ActivityNet (two datasets for which data is fully available). After that, we take the same model configurations and the same dataset but we train from scratch using our loss function. The results from this ablation study are reported in the following table:
>
> | Method | MSR-VTT | ActivityNet |
> |--------|---------|-------------|
> |        | R@1| R@1|
> | VAST   | 37.6    | 28.7        |
> | GRAM   | **38.9**    | **30.2**        |
>
> The proposed GRAM model significantly improves the alignment of the entire latent space thus leading to improved performance with respect to VAST and to conventional methods trained with cosine similarities. This approach not only enhances the alignment between text and video modalities but also outperforms the specialized TV loss. These results prove once again the increased learning and alignment performance of the proposed GRAM.
>
> [1] Miech, Antoine, et al. "Howto100m: Learning a text-video embedding by watching hundred million narrated video clips." IEEE/CVF ICCV 2019.
>
> [2] Zhu, Bin, et al. "Languagebind: Extending video-language pretraining to n-modality by language-based semantic alignment.", ICLR 2023.
>
> [3] Chen, Sihan, et al. "Vast: A vision-audio-subtitle-text omni-modality foundation model and dataset.", NeurIPS 2023.
>
> [4] Chen, Sihan, et al. "Valor: Vision-audio-language omni-perception pretraining model and dataset.", arXiv preprint arXiv:2304.08345, 2023.
>
> [5] Wang, Yi, et al. "Internvideo2: Scaling video foundation models for multimodal video understanding.", arXiv:2403.15377, 2024.
>
>
> **On the ablation studies.** According to the Reviewer’s suggestion, we conduct ablation experiments for all the losses we introduce in the paper $L_{D2A}$, $L_{A2D}$ and $L_{DAM}$.
> We report results in Appendix B.2, in Table 6 and Figure 5. We run ablation without our proposed losses (without D2A, A2D, and DAM), therefore adding only the standard cosine similarities (TV, TA), and combining the proposed losses.
>
> We report also here the table.
>
> |    |    |     |     |     | MSR-VTT (T2V) | MSR-VTT (V2T) | ActivityNet (T2V) | ActivityNet (V2T) |
> |----|----|-----|-----|-----|---------------|---------------|-------------------|-------------------|
> | TV | TA | D2A | A2D | DAM | T2V           | V2T           | T2V               | V2T               |
> | ✔  | ✔  | ✗   | ✗   | ✗   | 36.4          | 36.7          | 23.6              | 21.3              |
> | ✗  | ✗  | ✗   | ✔   | ✔   | 20.9          | 29.0          | 16.3              | 18.9              |
> | ✗  | ✗  | ✔   | ✗   | ✔   | 37.1          | 38.7          | 23.6              | 24.5              |
> | ✗  | ✗  | ✔   | ✔   | ✗   | 38.0          | 41.2          | 30.0              | 30.0              |
> | ✗  | ✗  | ✔   | ✔   | ✔   | **38.9**      | **41.9**      | **30.2**          | **30.1**          |
>
> From the Table, it is clear how combining the proposed $L_{D2A}$, $L_{A2D}$ and $L_{DAM}$ leads to the best performance both in MSR-VTT and in ActivityNet. In particular, the proposed combination of losses far exceeds the conventional configuration with cosine similarities by up to 6.6 R@1 points, once again proving the effectiveness of the proposed GRAM.

---

> > ### Comment · Reviewer_iJhF · 2024-11-24
> > **Thanks for the reply**
> >
> > I appreciate the author's detailed response. I will consider slightly increasing the score because some of my concerns are being relieved. By the way, I still have one concern:
> >  As you said that GRAM also allows the selection of diverse modalities beyond text, How much difference can be made by using different modalities as anchor？

---

> > > ### Author Response · Authors · 2024-11-26
> > > **Response to the additional question**
> > >
> > > We would like to sincerely thank the Reviewer for the positive response and for giving us the opportunity to further discuss this fundamental point.
> > >
> > > As the Reviewer correctly highlighted, the selection of the anchor in GRAM can be chosen *a priori* based on the intended goal. Following the literature, we selected the text as the anchor modality to validate GRAM. As demonstrated by numerous studies [1]-[5] (this is not an exhaustive list), the text proves to be a highly informative modality, rich in semantic content, and capable of easily describing multimodal contents.
> > > Moreover, the choice of the anchor modality is closely related to the downstream task. In our experiments, we perform text retrieval tasks such as V2T and T2V.
> > > To accomplish these tasks, it has been shown that designing a loss function with text as the anchor modality is a successful strategy [3]-[5]. Based on this knowledge, we followed the same strategy.
> > >
> > > In the scenario of T2V retrieval, choosing other anchor modalities besides text may not be ideal, but it could be instead beneficial if the downstream task changes.
> > > For instance, let us imagine that we want to accomplish Frames-to-Data retrieval, where Data is considered as an audio track with textual descriptions and possibly other modalities. In this case, the task is to assign a batch of frames to the correct data, or vice versa. The focus here has moved toward frames modality and so it would be advantageous to use frames as the anchor modality in both the training and inference phases.
> > >
> > > In order to further explore our statement, we perform zero-shot experiments on extracted embeddings from GRAM with different anchors. Without using the DAM module, since it is text-based from design, we show the raw results using Text as anchor (as in the main paper) and using video Frames as anchor. Of course when we use frames as anchor we are accomplished to a different task (i.e., Frames to text-audio, assigning a set of frames to the correct textual description/audio track couples). Results are shown in the following table:
> > >
> > > | Anchor Modality   | Task     | R@1   | R@1   | R@10   |
> > > |-------------------|----------|----------|----------|----------|
> > > | **Text Anchor**    | T 2 VA      | 46.49    | 74.89    | 84.39    |
> > > |   **Text Anchor**       | VA 2 T      | 45.20    | 73.44    | 83.15    |
> > > |||||
> > > | **Vision Anchor**  | Frames 2 TA      | 38.91    | 63.35    | 72.29    |
> > > |    **Vision Anchor**  | TA 2 Frames       | 43.55    | 72.06    | 82.58   |
> > >
> > > In conclusion, the choice of the anchor modality is **related to the downstream task**, since it has been proven that using anchors is beneficial to the final metrics. **Text** is, in general, considered a good anchor since it **well retains the broad semantics of multimodal data**.
> > >
> > > [1] Jacob Devlin Ming-Wei Chang Kenton and Lee Kristina Toutanova. Bert: Pre-training of deep bidirectional transformers for language understanding. In Proceedings of naacL-HLT, volume 1, pp. 2, 2019.
> > >
> > > [2] Swetha Sirnam, Mamshad Nayeem Rizve, Nina Shvetsova, Hilde Kuehne, Mubarak Shah, “Preserving Modality Structure Improves Multi-Modal Learning”, ICCV 2023.
> > >
> > > [3] Zhu, Bin, et al. "Languagebind: Extending video-language pretraining to n-modality by language-based semantic alignment." ICLR 2023.
> > >
> > > [4] Chen, Sihan, et al. "Vast: A vision-audio-subtitle-text omni-modality foundation model and dataset." Advances in Neural Information Processing Systems 36 (2023): 72842-72866.
> > >
> > > [5] Wang, Yi, et al. "Internvideo2: Scaling video foundation models for multimodal video understanding." arXiv e-prints (2024): arXiv-2403.

---

### Official Review · Reviewer_D5ua · 2024-11-03

**Soundness:** 3
**Presentation:** 3
**Contribution:** 3
**Rating:** 8
**Confidence:** 4

**Summary:**

This paper proposes a novel Gramian Representation Alignment Measure (GRAM) for multimodal alignment. Compared with the methods based on pairwise cosine similarity, GRAM scales well with the number of modalities.

**Strengths:**

1.The idea of taking advantage of determinant of the Gram matrix of the representation vectors of all modalities to measure their similarity is novel. Besides, the computational cost of the similarity calculation is low.

2.The proposed method provides a new insight to understand the representation geometric of multimodal learning.

**Weaknesses:**

1.The Gram model in the experiment section is obtained from VAST pretaining models, which make it difficult to verify that Gramian measure is superior to cosine similarity in aligning modalities. The reviewer suggests the authors train the model from scratch for a fair comparison.

2.Considering this Gramian measure is a new for modality alignment, the experimental analysis for it is not enough. The reviewer expects to observe how the Gramian similarity varies during training or fintuning, and compare it with cosine similarity.

**Questions:**

In table 2, all models except for UMT-L and InternVideo-L use 8 frames, and they two use 12 frames? Why adopt a unified setting for a fair comparison?

---

> ### Author Response · Authors · 2024-11-19
> **Response to Points 1 & 2 and Question 1**
>
> We would like to thank the Reviewer for the positive feedback and the valuable comments that helped us improve our paper and clarify several aspects.
>
> **On the training from scratch.** We thank the Reviewer for the insightful comment. The pretraining dataset of VAST is not fully available anymore due to the new policies of YouTube, so pretraining from scratch on a portion of the dataset would result in unfair results as well. However, following the Reviewer’s valuable suggestion, to better highlight the advantages of GRAM, we train it from scratch on MSR-VTT and ActivityNet, following the ablation strategies of VAST, so as to have fair comparisons. We report the results in Appendix B.2 in Figure 5 and in the following Table, where R@1 scores are reported.
>
> | Method | MSR-VTT | ActivityNet |
> |--------|---------|-------------|
> | VAST   | 37.6    | 28.7        |
> | GRAM   | **38.9**    | **30.2**        |
>
> The proposed GRAM model significantly improves the alignment of the entire latent space thus leading to improved performance with respect to VAST (in the Table) and to conventional methods trained with cosine similarities (in Fig.5 Appendix B.2).
> Therefore, in a fair training comparison from scratch on the same dataset GRAM outperforms the cosine-based VAST comparison, proving that the proposed GRAM-based loss functions are more effective in modeling the multimodal interactions with respect to the conventional cosine methods.
>
>
> **On the Gramian Value during training.** We would like to thank the Reviewer for the interesting suggestions. As suggested by the Reviewer, we plot the GRAM value and the cosine similarity during training. We report the plots in Figure 5 of Appendix B.2. From Fig.5, it is possible to see that:
>
> GRAM reduces the volume (gramian value) during training, while also indirectly increasing the cosine similarity (V2T R@1 and T2V R@1).
> GRAM increases the cosine similarity much more than the direct training with cosine similarity, meaning that GRAM builds overall a more aligned latent space.
> The cosine similarity training leaves almost unchanged the volume, meaning that it just works on two modalities and not on all the modalities.
> Therefore, while GRAM is able to reduce the volume, it also clearly improves the cosine similarity, more than the cosine similarity loss itself.
>
> **On the comparisons.** We select the results directly from previous studies since for the largest part of the models either there is not a working repository or checkpoints are not available. Therefore, since those models are finetuned with 12 frames, we just report those results. We run our experiments with 8 frames since it is the most common configuration, thus providing more meaningful and portable results.

---

> ### Comment · Reviewer_D5ua · 2024-11-22
> **Another question**
>
> Thank you for your response! My concerns are largely addressed. I have another question: Is it possible that the representations of all samples actually concentrate around a small area of the surface of the hyper ball (some kind of representation collapse)? If it is, how to prevent this happening.

---

> > ### Author Response · Authors · 2024-11-22
> > **Response to the additional question**
> >
> > We would like to sincerely thank the Reviewer for the positive response and the interesting point raised.
> >
> > We have investigated this aspect by computing the modality gap [1] of VAST and GRAM and we reported the results in Appendix B.3 Table 7 and in the following table.
> > Considering three modalities, Video (V), Text (T), and Audio (A), we compute the distances between the centroids of different modalities as suggested in [1] (e.g., VT → distance between the Vision centroid and the Text centroid, and so on).
> >
> >
> > | Method       | VT   | TA   | VA   |
> > |---------         |------|------|------|
> > | VAST ZS    | **0.65** | 0.84 | 0.96 |
> > | GRAM ZS   | 0.54 | **0.98** | **1.17** |
> > |                    |        |          |         |
> > | VAST FT     |0.37 | 0.86 | 0.84 |
> > | GRAM FT   |**0.40** | **0.96** | **1.09** |
> >
> >
> > As shown in the Table, the distances between the clusters are larger in GRAM, meaning that:
> >
> > * Latent representations of samples do not collapse on a small area of the hypersphere;
> >
> > * Samples in the latent space are much more spread around the hypersphere with respect to VAST (cosine-based method).
> >
> > Therefore, as far as we have observed, samples do not collapse on a small area of the hypersphere.
> >
> > [1] Mind the Gap: Understanding the Modality Gap in Multi-modal Contrastive Representation Learning. NeuIPS 2022.

---

> > > ### Comment · Reviewer_D5ua · 2024-11-23
> > > **score raised**
> > >
> > > Thank you for you effort in answering the question. I will raise the score.
> > >
> > > Good luck!

---

### Official Review · Reviewer_vQSX · 2024-11-04

**Soundness:** 3
**Presentation:** 3
**Contribution:** 3
**Rating:** 8
**Confidence:** 4

**Summary:**

The paper addresses limitations in conventional pairwise contrastive learning methods used for multimodal data integration, where alignment is typically achieved by mapping multiple modalities to a single anchor (e.g., image or text). These methods often fail to ensure semantic consistency among non-anchor modalities. The authors introduce a novel measure, the Gramian Representation Alignment Measure (GRAM), which utilizes the volume of a parallelotope formed by modality vectors in a higher-dimensional space to achieve a holistic alignment across modalities. The GRAM-based approach not only improves semantic alignment but also enhances performance on various multimodal tasks.

**Strengths:**

1、Innovative Methodology: The introduction of GRAM as a higher-dimensional geometric measure provides a new approach to aligning multiple modalities simultaneously, addressing the limitations of traditional pairwise methods.


2、Strong Empirical Results: The paper demonstrates substantial performance gains across multiple multimodal benchmarks, including video-audio-text retrieval and classification tasks, showcasing the effectiveness of the GRAM-based approach.

**Weaknesses:**

1. Does the volume complexity of GRAM computation increase significantly as the number of modalities increases? Especially in high dimensional space, does this calculation affect the real-time performance and scalability of the model? Has any consideration been given to approximate or optimize computations to reduce the computational burden?

2.GRAM is based on the volume of k-dimensional parallel polyhedra built from multimodal vectors as the alignment metric. For sparse or dense distributions in high-dimensional Spaces, does this volume still accurately reflect the semantic relations between modalities? Is there an experimental analysis to verify the robustness of this approach under different data distributions?

3.The paper mentions GRAM as an alternative to the traditional cosine similarity, but does this new alignment measure introduce instability or optimization difficulties during the actual training process? Have ablation experiments been performed to investigate the effect of GRAM on the convergence speed and stability of the model? In addition, are the effects of different learning rates and regularization strategies on model performance taken into account?

4.Does GRAM still provide a significant advantage in simple modal alignment tasks, or does its performance gain mostly manifest in complex multimodal interactions? Are there specific experimental comparisons that demonstrate the trade-off between GRAM and cosine similarity? In other words, under what circumstances is it advisable to choose GRAM over cosine similarity

**Questions:**

See Weaknesses.

---

> ### Author Response · Authors · 2024-11-19
> **Response to Points 1 & 2**
>
> We would like to thank the Reviewer for the positive feedback and the valuable comments that helped us improve our paper.
>
> **On the computation complexity.** We thank the Reviewer for giving us the opportunity to clarify this aspect. As we mentioned in Section 3.3, the volume computation **requires negligible computation time** as it relies on computing the determinant of a $k \times k$ matrix, where in the largest part of real-world situations $ k << n$, and often $k=3,4$.
> To further prove our claims, we empirically test cosine similarity computation and volume with the following setting:
>
> * Batchsize=256
>
> * n (latent space dimension)=512
>
> * k (number of modalities)=[2,3,4,5,10,20,30,40,50]
>
> The following Table shows the results.
>
> | Methods                    | 2        | 3        | 4        | 5        | 10       | 20       | 30       | 40       | 50       |
> |----------------------------|----------|----------|----------|----------|----------|----------|----------|----------|----------|
> | Pairwise cosine similarity | 1.29e-06 | 2.70e-06 | 3.90e-06 | 1.80e-06 | 1.17e-05 | 1.95e-05 | 5.52e-05 | 6.29e-05 | 6.66e-05 |
> | Gram Volume Computation    | 1.95e-05 | 2.73e-05 | 2.34e-05 | 3.90e-05 | 1.44e-04 | 7.64e-04 | 1.84e-03 | 2.69e-03 | 4.03e-03 |
>
> While the cosine similarity requires less time, volume computation is still fast and well-scale up to 50 modalities, each of dimension 512. Therefore, as we mentioned in Section 3.3, the additional computations are negligible. It is notable that in our experiments we run tests with 2,3,4, or 5 modalities, with the same batch size and latent dimension of this example, therefore our experiments require max 3.90e-05 seconds for the volume computation, which is quite good.
>
> **On the sparse and dense data distribution.** We thank the Reviewer for the interesting point raised.
> The Gram determinant depends on the relative orientation of the vectors, but it is not affected by their absolute density or sparsity [1]. Sparse distributions can result in high-dimensional vectors with many zeros, but the determinant remains valid as it is based on the linear independence and angles between the vectors. Similarly, dense distributions retain semantic relationships through their compact embeddings, ensuring the volume computation remains robust.
> Moreover, the model we test comprises transformer encoders (Sec. 4.1 Pag. 8). Usually, transformers learn contextualized and dense representations even from sparse or complex input data [2]. This ensures that the Gram determinant captures semantic relationships accurately, regardless of the initial distribution characteristics of the raw data.
>
> However, to further investigate the GRAM behavior with sparse data, we simulate a toy example. We randomly generate a normally-distributed vector and then we normalize it to norm 1, as the latent vectors in GRAM. Then, we masked this vector with different percentages in (40%, 60%, 80%). We expect that, coming from the same original distribution, these three masked vectors have low volume as the angle among them is small. We compute the determinant of the Gram matrix ($\mathbf{G}$) built with these three masked vectors, and the result is $\det(\mathbf{G})=0.092$, proving our hypothesis. Furthermore, if we sample two more distributions (one Poisson($\lambda=1$), one Gumbel($0, 1$)), and we compute the determinant of the Gram as:
>
> $$
> \mathbf{G} = \begin{bmatrix}
> N80 \cdot N80 & N80 \cdot P & N80 \cdot Gu \\\\
> P \cdot N & P \cdot P & P \cdot Gu \\\\
> Gu \cdot N & Gu \cdot P & Gu \cdot Gu
> \end{bmatrix}
> $$
>
> we expect that, given the sparse normal vector $N80$, the Poisson $P$, and the Gumbel $Gu$ coming from different distributions, the determinant of $\mathbf{G}$ is high. The result is $\det(\mathbf{G})=0.720$, once again proving that **the proposed method is stable and robust to dense and sparse data**.
>
> These results also prove that GRAM provides meaningful insights even in the case of very simple data and distributions and not only in the case of complex multimodal data.
>
> [1] Felix R. Gantmacher. Matrix theory. Chelsea Publishing Company, 1959.
>
> [2] Charles Zhang et al., “From Word Vectors to Multimodal Embeddings: Techniques, Applications, and Future Directions For Large Language Models”, ArXiv preprint: arXiv:arXiv:2411.05036, 2024.

---

> ### Author Response · Authors · 2024-11-19
> **Response to Points 3 & 4**
>
> **On the optimization and training of GRAM.** We would like to thank the Reviewer once again for giving us the opportunity to strengthen our claims. The determinant of the Gram matrix is computationally stable and does not introduce instability into the training process, as the Gram matrix, formed by computing the pairwise dot products of latent representations as formally defined in Eq. (3), is symmetric and positive semi-definite by construction. As a result, its determinant is well-defined and numerically stable, thus ensuring the effectiveness of the optimization process. To empirically prove our claims, we further performed ablation studies in Appendix B.2. Figure 5. Figure 5 shows a comparison between the same model trained with the classical cosine similarity losses (cosine between text and video TV and cosine between text and audio TA) and the proposed GRAM. As it is clear from Fig.5, especially from the loss function plot, the proposed GRAM:
>
> * obtains a **faster convergence**;
>
> * obtains a **better convergence**, as the loss is lower than the comparison;
>
> * has fewer variations, resulting in a **more regularized loss**.
>
> This proves that the introduced **GRAM loss functions are stable and improve the convergence** and optimization of models without requiring additional regularization strategies.
>
> **On the trade-off between GRAM and cosine similarity.** We thank the Reviewer for giving us the possibility to better investigate this point. Unlike cosine similarity, GRAM can be directly employed from 2 to $n$ modalities, therefore it can also directly replace the cosine similarity. Nevertheless, GRAM far surpasses cosine similarity approaches when each modality possesses semantic information that can significantly contribute to the correct discrimination in downstream tasks. A crucial example is provided in Appendix B.2, Figure 5, where we consider four triplets from YouTube consisting of video, audio, and text, and first compute the cosine similarity between text-video and text-audio pairs across all text labels and videos/audios. Then, we apply our proposed GRAM to the same dataset. As Figure 5 highlights, conventional cosine-based methods fail in jointly exploiting both audio and video modalities, often misleading when a single modality does not contain enough information for the correct classification. Conversely, the GRAM-based model exploits the semantically aligned multimodal space and jointly leverages all the modalities, leading to a better and correct retrieval in every case.

---

> > ### Comment · Reviewer_vQSX · 2024-12-03
> >
> > Thank you for the author's reply, most of my concerns have been resolved, so I will raise my score.

---

### Official Review · Reviewer_X14r · 2024-11-04

**Soundness:** 3
**Presentation:** 3
**Contribution:** 3
**Rating:** 6
**Confidence:** 4

**Summary:**

This manuscript proposed a novel strategy for the multimodal representation alignment, providing more informative semantics for multiple modalities through geometric alignment. The proposed method leverages Gramian volume to replace the traditional cosine similarity to preserve rich semantic Relations between multiple modalities. Additionally, the authors proposed the GRAM-based contrastive loss to improve the alignment between each modality. Finally,  they also proved the Gramian volume can serve as quantitative metrics to measure the performance of multimodal alignment for downstream tasks.  The paper is well-organized and provide a novel insight of multimodal learning and alignemnt.

**Strengths:**

1. The paper demonstrates a novel method to better align heterogeneous modalities in higher-dimensional geometric spaces, which take the sematic relation between multiple modalities into consideration.

2. It demonstrated that the proposed Gramian volume can be utlized as a novel metrics to evaluate the alignment performance of  the mulitmodal model.

3. The performance of the model is outstanding than other state-of-the-art methods.

4. The use of figures and tables in this paper to  illustrate the clear concepts and findings is effective.

**Weaknesses:**

1. The comparison on confusion matrices of cosine-based approach and proposed method should be exhibited in the former paper instead of in the supplementary materials. Also, in order to demonstrate the effectiveness of the proposed GRAM Multimodal Contrastive loss, it is important to conduct more ablation studies, comparing the vanilla contrastive loss. It is reasonable to conduct ablation studies including: (1) without proposed L_D2A and L_A2D, (2) without L_D2A, with L_A2D; (3) without L_A2D and with L_D2A; (4) without L_DAM .etc. on ActivityNet and VATEX  datasets. Also, it is more appropriate to exhibit the confusion matrix  in the supplimentary in the section 3.5 GRAM as Model Performance Metric to show the effectiveness of the proposed metrics.

2. In the introduction, the author should more explicity explain the concept of anchor modality. If the author quote this concept from another articles, please cite the articles.  Additionally, it is more plausible to conduct more explicit mathematic conduction on  superiority of the Gramm Volum comparing to trainditonal cosine similarity in the supplimentary. In the Figure 4, it is also neccessary to implement T-SNE visualization using cosine similarity to mask the comparisons.

3. Since the article proposed a new metrics,  it is neccessary to evalute other the-state-of-the-arts Methods (VAST, LanguageBind and VideoPrism-b and so on)  using the proposed metrcis.

4. In the table, the method's performances surpass the other the-state-of-the-art Methods significantly. In such case, the author should release the code link for the readers to validate after the publication or should provide anonymous link of the partial datasets in the table.

**Questions:**

1. The comparison on confusion matrices of cosine-based approach and proposed method should be exhibited in the former paper instead of in the supplementary materials. Also, in order to demonstrate the effectiveness of the proposed GRAM Multimodal Contrastive loss, it is important to conduct more ablation studies, comparing the vanilla contrastive loss. It is reasonable to conduct ablation studies including: (1) without proposed L_D2A and L_A2D, (2) without L_D2A, with L_A2D; (3) without L_A2D and with L_D2A; (4) without L_DAM .etc. on ActivityNet and VATEX  datasets. Also, it is more appropriate to exhibit the confusion matrix  in the supplimentary in the section 3.5 GRAM as Model Performance Metric to show the effectiveness of the proposed metrics.

2. In the introduction, the author should more explicity explain the concept of anchor modality. If the author quote this concept from another articles, please cite the articles.  Additionally, it is more plausible to conduct more explicit mathematic conduction on  superiority of the Gramm Volum comparing to trainditonal cosine similarity in the supplimentary. In the Figure 4, it is also neccessary to implement T-SNE visualization using cosine similarity to mask the comparisons.

3. Since the article proposed a new metrics,  it is neccessary to evalute other the-state-of-the-arts Methods (VAST, LanguageBind and VideoPrism-b and so on)  using the proposed metrcis.

4. In the table, the method's performances surpass the other the-state-of-the-art Methods significantly. In such case, the author should release the code link for the readers to validate after the publication or should provide anonymous link of the partial datasets in the table.

---

> ### Author Response · Authors · 2024-11-19
> **Response to Question 1**
>
> We would like to thank the Reviewer for the positive feedback and the valuable comments that helped us improve our paper and clarify several aspects.
>
> **On ablation studies.** According to the Reviewer’s suggestion, we conduct ablation experiments for all the losses we introduce in the paper $L_{D2A}$, $L_{A2D}$ and $L_{DAM}$. We report results in Appendix B.2, in Table 6 and Figure 5. We run ablation without our proposed losses (without D2A, A2D, and DAM), therefore adding only the standard cosine similarities (TV, TA), and combining the proposed losses.
>
> We report here Tab. 6 of Appendix B.2:
>
> |    |    |     |     |     | MSR-VTT (T2V) | MSR-VTT (V2T) | ActivityNet (T2V) | ActivityNet (V2T) |
> |----|----|-----|-----|-----|---------------|---------------|-------------------|-------------------|
> | TV | TA | D2A | A2D | DAM | T2V           | V2T           | T2V               | V2T               |
> | ✔  | ✔  | ✗   | ✗   | ✗   | 36.4          | 36.7          | 23.6              | 21.3              |
> | ✗  | ✗  | ✗   | ✔   | ✔   | 20.9          | 29.0          | 16.3              | 18.9              |
> | ✗  | ✗  | ✔   | ✗   | ✔   | 37.1          | 38.7          | 23.6              | 24.5              |
> | ✗  | ✗  | ✔   | ✔   | ✗   | 38.0          | 41.2          | 30.0              | 30.0              |
> | ✗  | ✗  | ✔   | ✔   | ✔   | **38.9**      | **41.9**      | **30.2**          | **30.1**          |
>
> From the Table, it is clear how combining the proposed $L_{D2A}$, $L_{A2D}$ and $L_{DAM}$ leads to the best performance both in MSR-VTT and in ActivityNet. In particular, the proposed combination of losses far exceeds the conventional configuration with cosine similarities by up to 6.6 R@1 points, once again proving the effectiveness of the proposed GRAM. We would like to thank the Reviewer once again for giving us this valuable suggestion that improved the clarity of our contributions.
>
> **On Fig.6.** We sincerely appreciate that the Reviewer noted the importance of Fig. 6, and we would like to thank her/him for this. This figure summarizes the impact of our work in real use cases.
> Unfortunately, the entire figure is too large to fit in the main paper and by reducing its size we would make it too confusing and consequently lose clarity. However, as suggested by the Reviewer, we have emphasized its importance by adding more references to it on Pag. 1 and Pag. 9.

---

> ### Author Response · Authors · 2024-11-19
> **Response to Question 2**
>
> **On the anchor.** We would like to thank the Reviewer for giving us the opportunity to clarify this point. We have updated the revised manuscript with the references introducing the concept of anchor/bridge modality. We report here the references and some explanations.
>
> [1] A. Jain, Y. Verma, “Cross-modal Retrieval Using Contrastive Learning of Visual-Semantic Embeddings”, ICPR 2022. > To the best of our knowledge, this paper introduces the concept of anchor modality.
>
> [2] Swetha Sirnam, Mamshad Nayeem Rizve, Nina Shvetsova, Hilde Kuehne, Mubarak Shah, “Preserving Modality Structure Improves Multi-Modal Learning”, ICCV 2023. > This paper proposes to model the relationship between samples with respect to their relationships with these anchors.
>
> [3] ImageBind paper >  Throughout the paper, the authors refer to the image modality as the “bridge” modality.
>
> **On the mathematic conduction.** We would like to thank again the Reviewer for her/his interesting suggestions. We developed the volume computation with GRAM for the three-modal case to mathematically show how GRAM better models the relationship between all the modalities and not only with the anchor one.
>
> Let us consider the simple case with three modalities: Text ($T$), Video ($V$), and Audio ($A$) with $T T, A A, V V$ dot products between the modalities themselves.
>
> The Gram Matrix is equal to:
>
> $$
> \mathbf{G} = \begin{bmatrix}
> T T & T A & T V \\\\
> A T & A A & A V \\\\
> V T & V A & V V
> \end{bmatrix}
> $$
>
> Now, let us compute the determinant of the matrix $\mathbf{G}$.
>
> $$
> \det(\mathbf{G}) = TT (AA \cdot VV - AV \cdot VA) - TA \cdot (AT \cdot VV - AV \cdot VT) + TV \cdot (AT \cdot VA - AA \cdot VT)
> $$
>
> Recall that $T,V,A$ have norm equal to $1$, so $TT = VV = AA = 1$:
>
> $$
> = 1 \cdot (1 - VA^2) - TA \cdot (AT - AV \cdot VT) + TV \cdot (AT \cdot VA - VT) \\
> = 1 - VA^2 - TA^2 + TA \cdot AV \cdot VT + TV \cdot AT \cdot VA - TV^2
> $$
>
> This results into:
>
> $$
> = 1 - VA^2 - TA^2 - TV^2 + 2 \cdot TA \cdot AV \cdot VT
> $$
>
> Therefore, the volume computation through the Gram matrix includes in its computation all the cross-products between the modalities ($ TA, VA, TV$), resulting in an alignment of all the modalities together. In contrast, current state-of-the-art methods based on cosine similarity only compute the similarities between the modalities and the anchor ($TA$ and $TV$), omitting the similarities between non-anchor modalities ($AV$), which in practice may not be aligned. The same computation can be expanded for $n$ modalities.
> Following the Reviewer’s advice, we added this explanation to Appendix A.2 in the revised manuscript.
>
>
> **On the tSNE plot.** In Figure 4 we plot the tSNE from VAST and GRAM. VAST is trained with cosine similarity, so the left plot in Figure 4 is already the output of a model trained with cosine similarity, resulting in a less aligned space with respect to the well-aligned GRAM space. We better clarify this aspect in the revised version manuscript and in the caption of the figure.

---

> ### Author Response · Authors · 2024-11-19
> **Response to Questions 3 and 4**
>
> **On the metric.** In this paper, we propose the Gramian Representation and Alignment Measure (GRAM), which is a measure we include in novel contrastive learning loss functions to improve the multimodal alignment and as a metric to evaluate large multimodal models. We evaluate the proposal of GRAM as a metric for model performance in Sec. 3.5 on LanguageBind, VAST, and the proposed GRAM Model. Following the Reviewer’s advice, we have updated the evaluation and the plot. The largest part of comparison methods is two-modal, only comprising text and video (like VideoPrism), and for these cases, the GRAM computation relies on the angle between the two modalities vector, thus similar to cosine similarity. Therefore we do not evaluate GRAM on them, as no further insights can be gained from them. Several three- or four-modal models do not have a working repository or available checkpoints. Therefore, we compute the GRAM metric on VALOR, which released both working code and checkpoints. We have updated Figure 3 in Sec. 3.5 with the VALOR evaluation. Interestingly, also VALOR confirms our claims that the models’ performance is strictly correlated ($\rho=0.923$) to the GRAM metric.
> We would like to thank the Reviewer once again for giving us this valuable suggestion.
>
> **On the code.** We have already attached the code to reproduce GRAM experiments in zero-shot scenarios as supplementary material to the initial submission. Of course, we plan to release the full repository including all the checkpoints (pretraining, finetuning, from scratch) after the review process, as we are still refining all the code. Furthermore, we would like to thank the Reviewer for the suggestion of also releasing the partial datasets for further investigation. To this end, we further attached the files corresponding to the video IDs of our partial datasets to the supplementary.
>
> The Reviewer can visit the anonymized repository (still to be refined) at the link: https://anonymous.4open.science/r/GRAM-681D/README.md

---

### Author Response · Authors · 2024-11-19
**Summary of changes**

We would like to sincerely thank all the Reviewers for their insightful comments and suggestions, we believe all their advice helped us improve the paper.

To summarize, we list here a summary of the main changes we made following Reviewers' suggestions:

* We performed **additional ablation studies** on the proposed loss functions. We report all the results in Appendix B.2.

* We **trained from scratch** a cosine-based model and GRAM on two datasets to provide fair comparisons with equal models, hyperparameters, and datasets on MSR-VTT and ActivityNet. The losses and curves are shown in Fig. 5 Appendix B.2.

* We directly **compared and studied** how the Gramian Value and the cosine similarity vary during training with conventional methods and with GRAM, results in Appendix B.2.

* We performed a **study on the modality gap** in GRAM and we show the conclusions and the results in Appendix B.3.

* We **mathematically proved the theoretical advantages** of the proposed volume computation with respect to cosine similarity in the multimodal setting. The proof is available in Appendix A.2.

* We **added more comparison methods** for proving that the GRAM metric is strongly correlated with models’ performance. Results are shown in Fig. 3.

* We **emphasized the importance of Figure 6** comprising real use cases in which conventional cosine-based methods fail while GRAM successes by adding more references to it in the main text of the paper.

* We better **specified the way to properly select the anchor** modality in GRAM.

---

### Author Response · Authors · 2024-12-01
**Discussion period ending, we would appreciate any further feedback**

Dear Reviewers,

The deadline for the discussion phase is approaching and we believe we have addressed the raised concerns in the rebuttal.

We would like to thank the Reviewers once again for their positive feedback and constructive comments. We would appreciate it if you could parse our responses and let us know if there are any further questions we can answer to clarify any doubts.

---

### Meta-Review · Area_Chair_SxZG · 2024-12-16

**Metareview:**

The paper presents an approach to multimodal learning that addresses the limitations of traditional pairwise alignment methods, which often fail to ensure proper integration of multiple modalities. The authors propose the Gramian Representation Alignment Measure (GRAM), which minimizes the Gramian volume of the parallelotope spanned by modality vectors to achieve simultaneous alignment of all modalities. GRAM enhances existing contrastive loss functions, replacing cosine similarity and improving performance in tasks like video-audio-text retrieval and audio-video classification.

The reviewers were nearly unanimous in recognizing the following strengths of the paper:

+ The novelty of using the Gramian volume as a geometric measure to characterize multi-modal alignment.
+ The very strong empirical results demonstrating superior performance on a range of multi-modal benchmarks when using GRAM.
+ The excellent insights on the geometry of multi-modal learning, which are very well supported by the paper motivations and illustrative examples.

In the initial reviews, reviewers questioned the following points:

+ The fairness of the experimental setup given that the proposed model was pre-trained on new data using the proposed loss, while the compared methods were not. In the rebuttal the authors addressed this point with additional experiments in a controlled setting comparing VAST with GRAM on MST-VTT and ActivityNet (two publicly available datasets). This additional experiment, along with the new ablations, adequately address concerns about fairness.
+ Potential inefficiency and/or instability when using GRAM as an alignment measure. In rebuttal the authors also addressed this issue with new details regarding the efficiency of the Gramian volume computation, as well as new ablations demonstrating the stability of GRAM during training.

During the author/reviewer discussion period there were a number of exchanges between three of the reviewers and the authors. The author rebuttals were convincing in these cases, adequately addressing all major concerns. A strong consensus towards Accept emerged in the end.

**Additional Comments On Reviewer Discussion:**

In the original reviews, two reviewers asked direct and pointed questions about the stability of GRAM as a metric during training, and about potential problems with fairness due to different pre-training data used by models in the comparative analysis. The author rebuttals, including additional details and a few new ablations, clarified these points to the satisfaction of the reviewers (and the AC).

---

### Decision · Program_Chairs · 2025-01-22

Accept (Poster)